# Temporal structure of natural language processing in the human brain corresponds to layered hierarchy of large language models

Ariel Goldstein [1,2,3,12] ✉, Eric Ham[4,5,12], Mariano Schain[3,12], Samuel A. Nastase [4], Bobbi Aubrey[4,6], Zaid Zada [4], Avigail Grinstein-Dabush[3], Harshvardhan Gazula[4], Amir Feder[3], Werner Doyle[6], Sasha Devore[6], Patricia Dugan[6], Daniel Friedman[6], Michael Brenner [3,7], Avinatan Hassidim[3], Yossi Matias [3], Orrin Devinsky [6], Noam Siegelman [1,8], Adeen Flinker[6,9], Omer Levy[10], Roi Reichart[11] & Uri Hasson [3,4]

Large Language Models (LLMs) offer a framework for understanding language processing in the human brain. Unlike traditional models, LLMs represent words and context through layered numerical embeddings. Here, we demonstrate that LLMs' layer hierarchy aligns with the temporal dynamics of language comprehension in the brain. Using electrocorticography (ECoG) data from participants listening to a 30-minute narrative, we show that deeper LLM layers correspond to later brain activity, particularly in Broca's area and other language-related regions. We extract contextual embeddings from GPT-2 XL and Llama-2 and use linear models to predict neural responses across time. Our results reveal a strong correlation between model depth and the brain's temporal receptive window during comprehension. We also compare LLM-based predictions with symbolic approaches, highlighting the advantages of deep learning models in capturing brain dynamics. We release our aligned neural and linguistic dataset as a public benchmark to test competing theories of language processing.

Large Language Models (LLMs) provide an alternative computational framework for understanding how the human brain processes natural language[1–5]. Classical psycholinguistic models rely on rule-based manipulation of symbolic representations embedded in hierarchical tree structures[6,7]. In sharp contrast, LLMs encode words and their context as continuous numerical vectors–i.e., embeddings. These embeddings are constructed via a sequence of nonlinear transformations across layers to yield the sophisticated representations of linguistic structures needed to produce language[8–12]. These layered models and their resulting representations enable many emerging

[1]Department of Cognitive and Brain Sciences, Hebrew University, Jerusalem, Israel. [2]Business School, Hebrew University, Jerusalem, Israel. [3]Google Research, Tel-Aviv, Israel. [4]Department of Psychology and the Neuroscience Institute, Princeton University, Princeton, NJ, USA. [5]Bioinformatics Interdepartmental Program, University of California, Los Angeles, Los Angeles, CA, USA. [6]New York University Grossman School of Medicine, New York, NY, USA. [7]School of Engineering and Applied Science, Harvard University, Cambridge, MA, USA. [8]Department of Psychology, Hebrew University, Jerusalem, Israel. [9]New York University Tandon School of Engineering, Brooklyn, NY, USA. [10]Blavatnik School of Computer Science, Tel-Aviv University, Tel-Aviv, Israel. [11]Technion—Israel Institute of Technology, Haifa, Israel. [12]These authors contributed equally: Ariel Goldstein, Eric Ham, Mariano Schain. ✉e-mail: ariel.y.goldstein@mail.huji.ac.il

applications, such as language translation and human-like text generation.

LLMs embody three fundamental principles for language processing: (1) embedding-based contextual representation of words, (2) next-word prediction, and (3) error correction-based learning. Recent research has begun identifying neural correlates of these computational principles in the human brain as it processes natural language. First, contextual embeddings derived from LLMs provide a powerful model for predicting the neural response during natural language processing[1,2,4,13,14]. For example, neural responses recorded in the Inferior Frontal Gyrus (IFG) seem to align with contextual embeddings derived from a specific LLM (i.e., GPT2-XL)[1,2,13]. Second, spontaneous pre-word-onset next-word predictions were found in the human brain using electrophysiology and imaging during free speech comprehension[1,15,16]. In addition, these predictions aligned with GPT2-XL predictions[1,16]. Third, an increase in post-word-onset neural activity for surprising words has been reported in language areas[1,17,18]. Likewise, for surprising words, linear encoding models trained to predict brain activity from word embeddings have also shown higher performance hundreds of milliseconds after word onset in higher-order language areas[1], suggesting an error response during human language comprehension. These results highlight the potential for LLMs as cognitive models of human language comprehension[1–4].

Following these fundamental shared principles, we explore whether the sequential-layered process implemented by the LLMs is also shared with the neural temporal dynamic of comprehension in the human brain. In other words, is the hierarchy induced by the LLMs similar to the brain's temporal dynamic when listening to free speech? Our work leverages the superior spatiotemporal resolution of electrocorticography (ECoG,[19,20]) to show that the human brain's internal temporal processing of spoken narrative matches the internal sequence of nonlinear layer-wise transformations in LLMs. Specifically, in our study, we used ECoG to record neural activity in language areas along the superior temporal gyrus and inferior frontal gyrus (IFG) while human participants listened to a 30-min spoken narrative. We supplied this same narrative to high-performing LLMs[8,21] and extracted the contextual embeddings for each word in the story across all layers of the models. Following prior work, we use GPT2-XL to demonstrate the fundamental similarities between LLMs and the brain[1,2,4,13]. In addition, we take it one step further and generalize the result to the state-of-the-art open-source LLM Llama 2[21], supporting the argument that there are key similarities between the way LLMs and the brain process language.

We mapped the internal sequence of embeddings across the layers of LLMs to the neural responses recorded via ECoG in the human participants during natural language comprehension. To perform this comparison, we measured the performance of linear encoding models trained to predict temporally evolving neural activity from the embeddings at each layer. Our performance metric is the correlation between the true neural signal and the neural signal predicted by our encoding models. Specifically, within the inferior frontal gyrus (IFG), we observed a temporal sequence in our encoding results where earlier layers yield peak encoding performance near word onset, and later layers yield peak encoding performance later in time. This finding suggests that the transformation sequence across layers in LLMs maps onto a temporal dynamic of information processing in high-level language areas. In other words, we found that the spatial, layered hierarchy of LLMs may be used to model the temporal dynamics of language comprehension. We subsequently applied this analysis to other language areas along the linguistic processing hierarchy. We validated existing work that suggested an accumulation of information over increasing time scales, moving up the hierarchy from auditory to syntactic and semantic areas[22].

We replicated the finding that intermediate layers best predict cortical activity[23,24]. However, the improved temporal resolution of our ECoG recordings revealed an alignment between the layer-wise

sequence of LLM embeddings and the temporal dynamics of cortical activity during natural language comprehension.

All in all, this study provides evidence of the shared computational principles between LLMs and the human brain by demonstrating a connection between the layer-wise sequence of computations in LLMs and human brain electrical activity during natural language processing. We explore the progression of nonlinear transformations of word embeddings through the layers of deep language models and investigate how these transformations correspond to the human brain's hierarchical processing of natural language (Fig. 1).

The main contributions of this paper are the following: First, we provide the first evidence (to the best of our knowledge) that the layered hierarchy of LLMs like GPT2-XL and Llama 2 can be used to model the temporal hierarchy of language comprehension in a high-order human language area (Broca's Area: Fig. 2A). This suggests that the computations done by the brain over time during language comprehension can be modeled by the layer-wise progression of the computations done in the LLMs. Second, we validate our model by applying it to other language-related brain areas (Fig. 3). These analyses replicate neuroscientific results that suggest an accumulation of information along the linguistic processing hierarchy and, therefore, show LLMs' promise as models of the temporal hierarchy within human language areas and the spatial hierarchy of these areas in the human brain. Third, we use the same dataset to test the performance of a classical-symbolic approach to predict the temporal dynamic of the brain. We used symbolic representations for phonemes, morphemes, syntax, and semantics and evaluated their ability to predict the temporal dynamic. The results showed the superiority of LLMs and open the door to further comparisons with other theories. Finally, we provide the dataset yielding the results above. We offer this dataset as a benchmark for testing existing and future theories for the temporal dynamics of neural activity related to free speech comprehension.

## Results

We collected electrocorticography (ECoG) data from 9 epilepsy patients (7 of them had electrodes in the pre-defined ROIs) while they listened to a 30-min audio podcast ("Monkey in the Middle", NPR 2017), and preprocessed the neural data to reflect the high-gamma band signal (see Supplementary Fig. 11). In prior work, embeddings were taken from the final hidden layer of GPT2-XL to predict brain activity, and it was found that these contextual embeddings outperform static (i.e., non-contextual) embeddings[1,4,24]. In this paper, we expanded upon this analysis by modeling the neural responses for each word in the podcast using contextual embeddings extracted from all hidden layers of GPT2-XL and Llama-2. We focused on four areas along the ventral language processing stream[25–27]: middle Superior Temporal Gyrus (mSTG) with 28 electrodes, anterior Superior Temporal Gyrus (aSTG) with 13 electrodes, Inferior Frontal Gyrus (IFG) with 46 electrodes, and the Temporal Pole (TP) with 6 electrodes. We selected electrodes previously shown to have significant encoding performance for static (GloVe) embeddings (corrected for multiple comparisons). Finally, given that prior studies have reported improved encoding results for words correctly predicted by LLMs[1,3], we separately modeled the neural responses for correct predictions and incorrect predictions. We considered predictions correct if GPT2-XL assigned the correct word with the highest probability of coming next. There were 1709 of these words in the podcast, which we called the top 1 predicted or just predicted words. We also contrasted these encoding results with those for words that GPT2-XL was not able to predict. To match statistical power across the two analyses, we defined incorrect predictions as cases where the correct word was not among the top 5 most probable next words as determined by GPT2-XL, which gave us 1808 incorrectly predicted words. We denote these top 5 not predicted or just not predicted words. In Supplementary Figs. 2, 3 and 4, we ran our analysis for correct, incorrect and all words. To

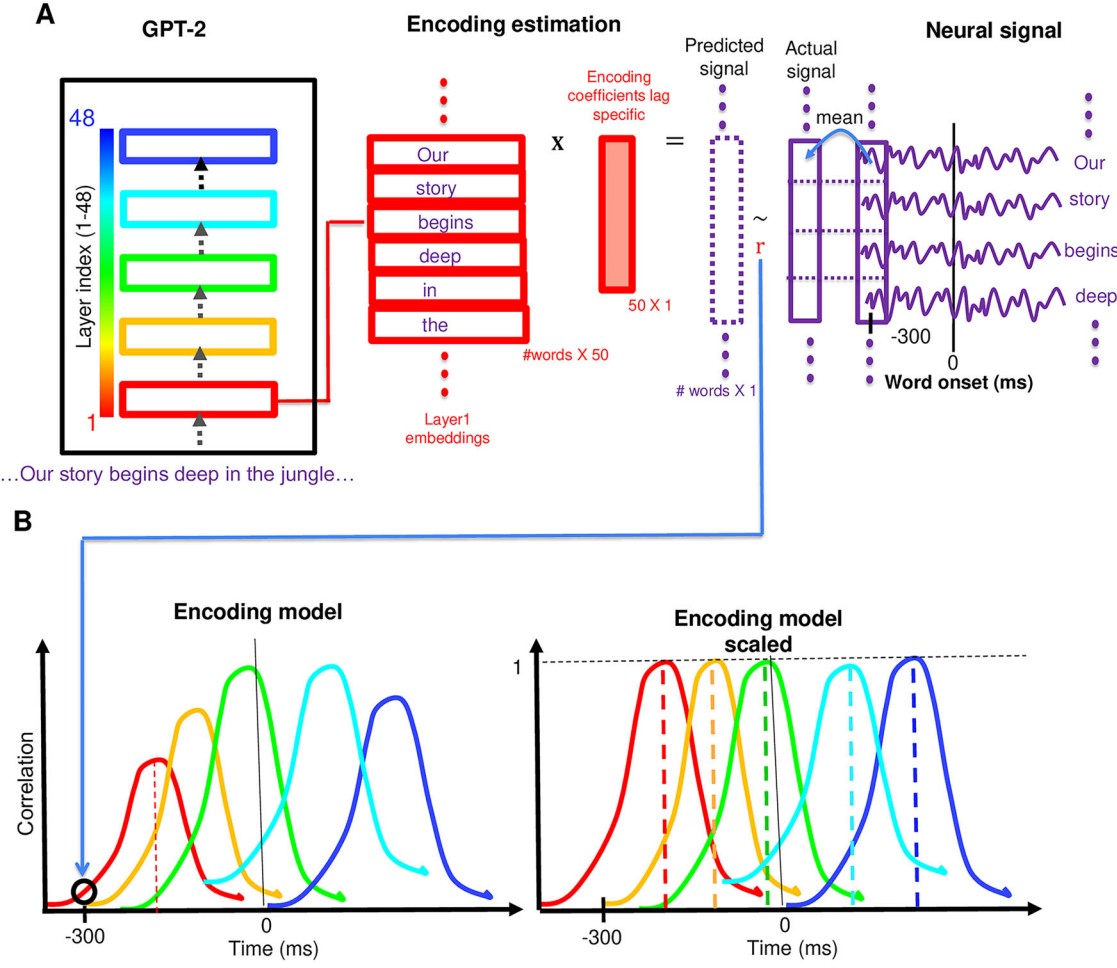

**Fig. 1 | Layer-wise encoding models. A** Estimating encoding models for each combination of electrode, lag, and layer: The word embedding for an input word is extracted from a designated layer of the LLM. A linear model is trained to estimate the lagged brain signal at an electrode from the word embedding. We color-coded the encoding performance according to the index of the layer from which the embeddings were extracted (red-blue scale), ranging from layer 1 (red) to layer 48 (blue). **B** left Illustration of encoding performance plot. **B** right Illustration of scaled encoding performance plot.

establish the robustness of the claim to LLMs in general, we reproduced the results using Llama 2 (Supplementary Figs. 5 and 7).

## Encoding model

For each electrode, we extracted 4000 ms windows of neural signal around each word's onset (denoted lag 0). The neural signal was averaged over a 200 ms rolling window with incremental shifts of 25 ms. The words and their corresponding neural signals were split into 10 non-overlapping subsets for a tenfold cross-validation training and testing procedure. For each word in the story, a contextual embedding was extracted from the output of each layer of the LLM (for example, Fig. 1; layer 1: red). The dimension of the embeddings was reduced to 50 using PCA. We performed PCA per layer to avoid mixing information between the layers. For each combination of electrode, layer, and lag relative to word onset, we ran a separate linear regression to estimate an encoding model that takes that layer's word embeddings as input and predicts the corresponding neural signals recorded by that electrode at that lag relative to word onset. We estimated the linear model from each 9-training folds and evaluated it on the held-out test-fold. We repeated this for all folds to get predictions for all words at that lag. We evaluated the model's performance by computing the correlation between the predicted and true values for all words (Fig. 1A). This process was repeated for all electrodes, for lags ranging from −2000 ms to +2000 ms (in 25ms increments) relative to word onset, and using the embeddings from each of the layers of the

LLM: 48 for GPT2-XL and 32 for Llama 2 (Fig. 1B). For GPT2-XL, the result for each electrode is a 48 by 161 matrix of correlations, where we denote each row (corresponding to all lags for one layer) as an encoding. We color-coded the encoding performance according to the index of the layer from which the embeddings were extracted, ranging from 1 (red) to 48 (blue), see Fig. 1A. We averaged the encodings over electrodes in the relevant ROIs to evaluate our procedure on specific ROIs. We then scaled the encoding model performance for each layer such that it peaks at 1; this allowed us to more easily visualize the temporal dynamics of encoding performance across layers.

## LLMs as a model for the temporal hierarchy of language comprehension in the Inferior Frontal Gyrus

We started by using GPT2-XL embeddings and focusing on neural responses for correctly predicted words in electrodes at the inferior frontal gyrus (IFG), a central region for semantic and syntactic linguistic processing[1,5,28–32].

For each electrode in the IFG, we performed an encoding analysis for each GPT2-XL layer (1–48) at each lag (−2000 ms to 2000 ms in 25 ms increments). We then averaged encoding performance across all electrodes in the IFG to get a single mean encoding time course for each layer (all layers are plotted in Fig. 2C). We averaged over electrodes and lags to get an average encoding performance per layer (Fig. 2B). Significance was assessed using bootstrap resampling across electrodes (see statistical significance in Method). The peak average

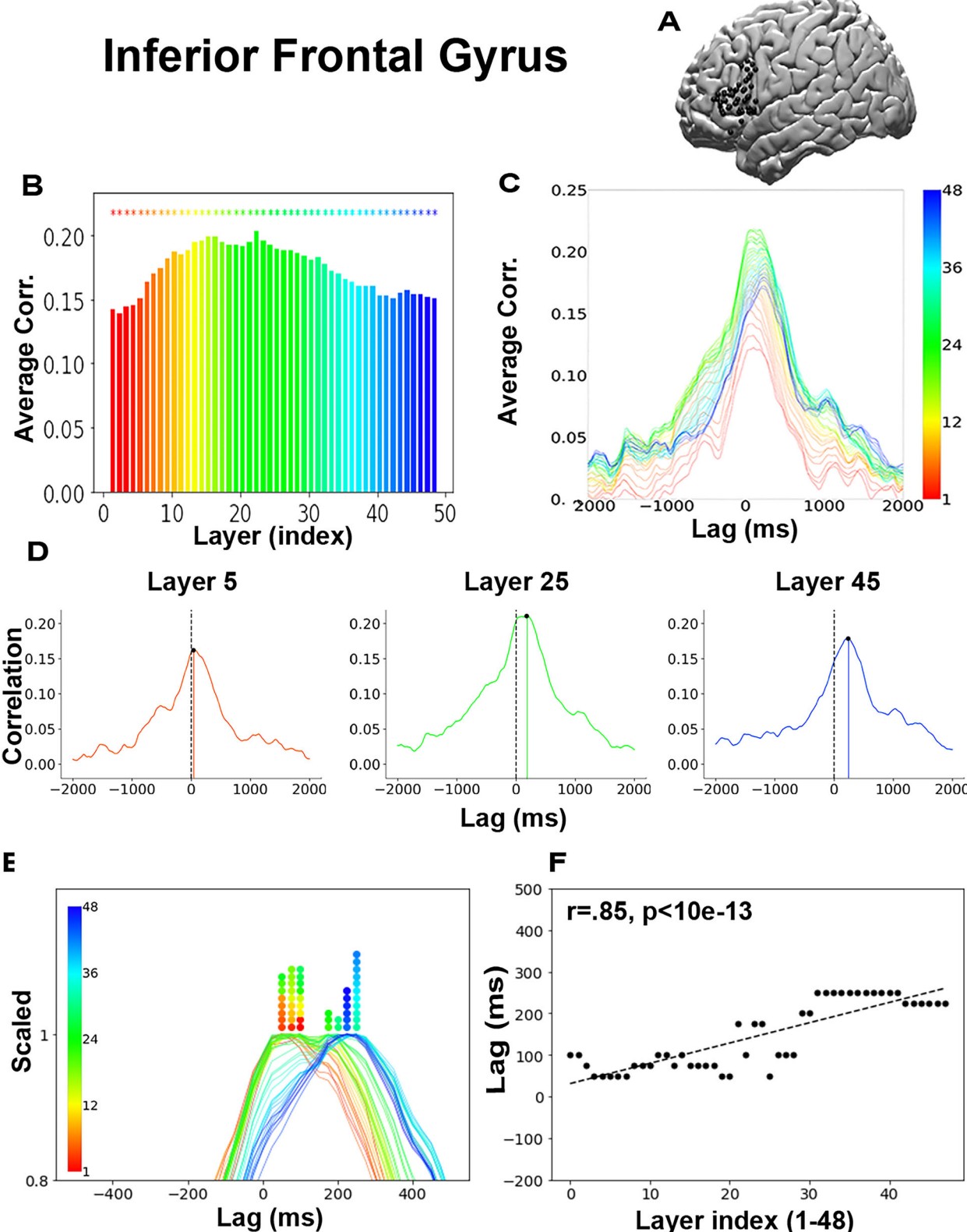

Fig. 2 | **Temporal dynamics of layer-wise encoding for correctly predicted words in IFG. A** Location of IFG electrodes on the brain (black). **B** Average max-encoding performance across IFG electrodes for each layer. Color coded by index (red-blue) as described in Fig. 1. **C** Per-layer encoding plot in the IFG. **D** Encoding performance for layers 5, 25, and 45 shows the layer-wise shift of peak performance across lags. **E** Scaled encodings. **F** Scatter plot of the lag that yields peak encoding performance for each layer. A one-sided $p$-value for the lag-layer correlation is reported ($N = 48$).

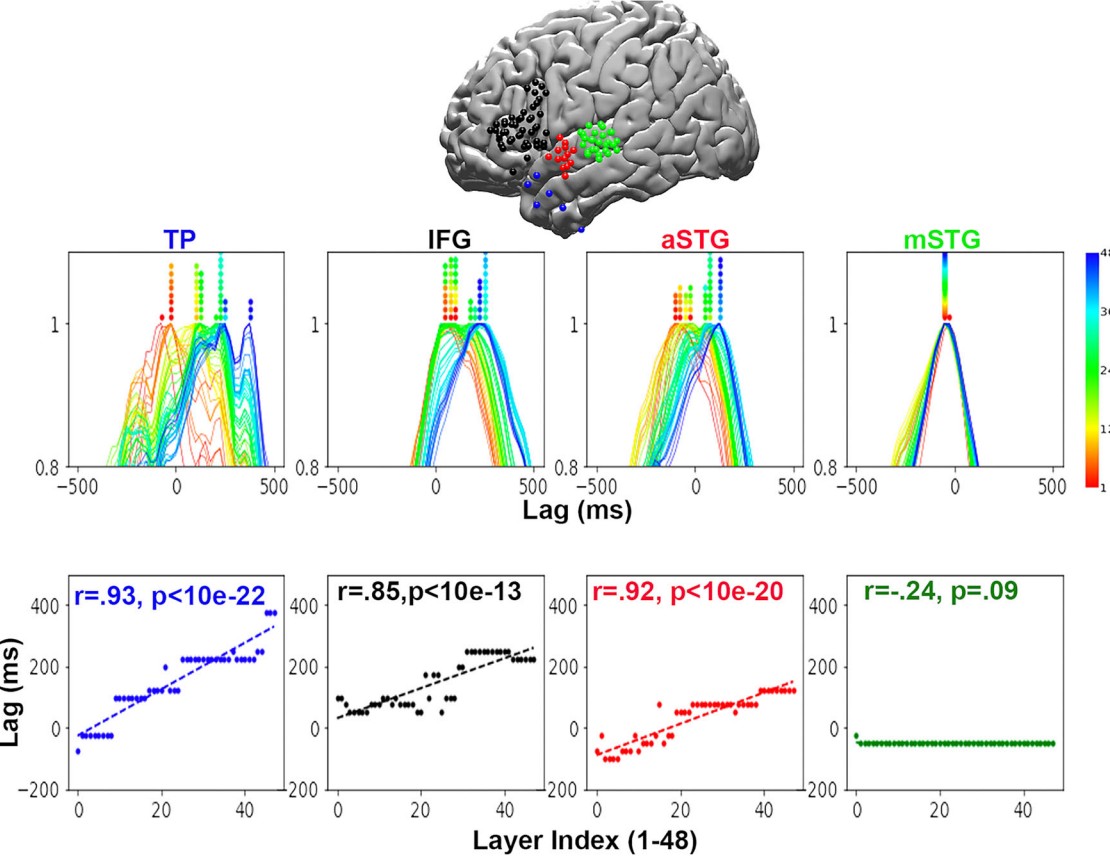

**Fig. 3 | Temporal hierarchy along the ventral language stream for correctly predicted words.** (Top) Location of electrodes on the brain, color-coded by ROI with blue, black, red, and green corresponding to TP, IFG, aSTG, and mSTG, respectively. (Middle) Scaled encoding performance for these ROIs. Color coded by layer index (red-blue) as described in Fig. 1. (Bottom) Scatter plot of the lag that yields peak encoding performance for each layer. One-sided $p$ values for the lag-layer correlation are reported ($N = 48$).

correlation of the encoding models in the IFG was observed for the intermediate layer 22 (Fig. 2B; for other ROIs and predictability conditions, see Supp. Figs. 2, 3 and 4). This corroborated recent findings from fMRI[4,24,33] where encoding performance peaked in the intermediate layers, yielding an inverted U-shaped curve across layers (Fig. 2B, Supplementary Fig. 1). The fine-grained temporal resolution of ECoG recordings suggested a specific dynamic pattern. All 48 layers yielded robust encoding in the IFG, with encoding performance near zero at the edges of the 4000 ms window and increased performance around word onset. This can be seen in the combined plot of all 48 layers (Fig. C; for other ROIs and predictability conditions, see Supplementary Fig. 3) and when we plotted individually selected layers (Fig. 2D, layers 5, 25, 45). A closer look at the encoding results over lags for each layer revealed an orderly dynamic in which the peak encoding performance for the early layers (e.g., layer 5, red, in Fig. 2D) tended to precede the peak encoding performance for intermediate layers (e.g., layer 25, green), which were followed by the later layers (e.g., layer 45, blue). To visualize the temporal sequence across lags, we normalized the encoding performance for each layer by scaling its peak performance to 1 (Fig. 2E; for other ROIs and predictability conditions, see Supplementary Fig. 4). To quantitatively test this claim, we computed the Pearson correlation between the layer index (1–48) and the lag that yielded the peak correlation for that layer (Fig. 2F). This yielded a strong, significant positive Pearson correlation of 0.85 (with two-sided $p$ value, $p < 10e{-}13$, obtained using the exact distribution of Pearson R). We call the result of this procedure the lag-layer correlation when using Pearson correlation. Similar results were obtained with Spearman correlation; $r = 0.80$. We also conducted a non-parametric analysis where we permuted the layer index 100,000 times (keeping the

lags that yielded the peak correlations fixed) and correlated the lags with these shuffled layer indices. Using this null distribution of correlations, we computed the percentile of the actual correlation ($r=0.85$) and got a significance of $p < 10e{-}5$. We replicated the results with Llama-2 (Supplementary Fig. 5) supporting the claim for general similarity between the hierarchy induced by LLMs and the human brain.

While, while we observed a robust correspondence between layer index and the lag of that layer's peak correlation, some groups of layers reached maximum correlations at the same lag. These nonlinearities could be due to discontinuity in the match between GPT2-XL's 48 layers and transitions within the individual language areas. Alternatively, this may be due to the temporal resolution of our ECoG measurements, which, although high, were binned at 50 ms resolution. In other words, it is possible that higher-resolution ECoG data would disambiguate these layers.

We and others observed that the middle layers in LLMs better fit the neural signals than the early or late layers. Following the observation that embeddings from intermediate layers consistently outperform those from other layers across lags, we conducted a regression analysis to determine if the source of the correlations between brain activity and early and late layer embeddings was due to similarities with the 'optimal' intermediate layer (layer 22). Our analysis revealed that the significant correlations with early and late layers are due to unique information that they independently encode (Supplementary Fig. 6; for a full description of the procedure, see Section 11). This finding underscores each layer's embeddings' independent and distinct contributions to our understanding of neural dynamics in language comprehension.

To determine the significance of spatial differences in timing across subregions of the IFG, we conducted a statistical comparison of peak lag times between BA44 (M = 155 ms, SD = 95) and BA45 (M = 152 ms, SD = 72), the two subregions for which we had sufficient electrode coverage (Supplementary Fig. 12). A paired-sample $t$ test revealed no significant difference in timing (t(47) = 0.26, $p$ = 0.79). A Bayesian paired-sample t-test yielded a Bayes factor of 6, indicating moderate evidence for the null hypothesis and suggesting that the data do not support a meaningful difference in timing between these subregions.

Together, these results suggested that for correct predictions the sequence of internal transformations across the layers in GPT2-XL matches the dynamics of neural transformations across time within the IFG. We replicated the results for Llama 2 (Supplementary Fig. 5), supporting the claim that LLMs generally model the process in the human brain during comprehension.

### Using LLMs to recover the increasing temporal receptive window along the linguistic processing hierarchy

We compared the temporal encoding sequence across three additional language ROIs (Fig. 3; replication for Llama 2 at Supplementary Fig. 7), starting with mSTG (near the early auditory cortex) and moving up along the ventral linguistic stream to the aSTG and TP (Due to the electrode placements, we did not have electrodes in the pSTS (see Supp. Table 1 for the results per-participant per-ROI). We did not observe obvious evidence for a temporal structure in the mSTG (Fig. 3, bottom right). This suggested that the temporal dynamic observed in the IFG is regionally specific and does not occur in the early stages of the language processing hierarchy. In addition to the IFG, we found evidence for the same orderly temporal dynamic in the aSTG (Pearson r = 0.92, $p$ value, $p < 10e−20$) and TP ($r = 0.93$, $p < 10e−22$). Similar results were obtained with Spearman correlation (mSTG r = −0.24, $p$ = 0.09; aSTG r = 0.94, $p < 10e−21$; IFG r = 0.80, $p < 10e−11$; TP r = 0.96, $p < 10e−27$), demonstrating that the effect is robust to outliers. We followed our procedure for the IFG and conducted permutation tests by correlating 100,000 sets of permuted layer indices with the true lags of peak encoding correlation for each layer. The resulting $p$ values were p < 0.02 for the mSTG, and $p < 10e−5$ for the aSTG and IFG. We ran a linear mixed model for the relationship between layer order and latency across ROIs, structured to generalize across electrodes and participants (Model: lag ~ 1 + layer + ROI + (1 + layer | electrode) + (1+layer | participant)). All fixed effects in this model were significant ($p < 0.001$), indicating that the influence of layer order on latency is robust, extending across different electrodes and patients.

Our results suggest that neural activity in language areas proceeds through a nonlinear set of transformations that match the nonlinear transformations in deep language models. An alternative hypothesis is that the lag-layer correlation is due to a more rudimentary network property, in which early layers represent the previous word, late layers represent the current word, and intermediate layers carry a linear combination of both words. To test this alternative explanation, we designed a control analysis where we generated 10,000 sets of 46 intermediate pseudo-layers by linearly interpolating between the first and last layers' embeddings. We computed the lag-layer correlation for each set (see 11). Supp. Fig. 8 plots the slopes obtained for the controlled linear transformations versus the actual nonlinear transformations. The results indicate that the actual lag-layered correlations were significantly higher than the ones achieved by the linearly-interpolated layers ($p < 0.01$). This indicates that GPT2-XL, with its non-linear transformations captured the brain dynamics better than a simpler model that performed a linear transformation between the embeddings of the previous and current words.

We then explored the timescales of the temporal progression in different brain areas. It seemed that the timescales gradually increased along the ventral linguistic hierarchy (see the steepness of the slopes across language areas in Fig. 3). For each ROI, we computed standard deviations of the set of encoding maximizing lags as a proxy for timescale. We used Levene's test and found significant differences between these standard deviations for the mSTG and aSTG (F = 48.1, $p < 0.01$) and for the aSTG and TP (F = 5.8, $p < 0.02$). The largest within-ROI temporal separation across layer-based encoding models was seen in the TP, with more than a 500 ms difference between the peak for layer 1 (around −100 ms) and the peak for layer 48 (around 400 ms).

### An alternative psycholinguistics approach

Traditionally, neurolinguistics employs a systematic approach that breaks down the human language into hierarchical representations of phonology, morphology, syntax, and semantics[34,35]. Each layer of representation consists of specific symbolic features. Although there are known interactions among these levels of linguistic representation, research labs have typically focused on studying these components separately, conducting experiments tailored to individual subfields[36–39]. In our study, we developed vectorial representations for each level within this psycholinguistic hierarchy used this framework to produce psycholinguistic embeddings for the benchmark dataset.

Phonemes are the smallest units of distinguishable sounds in a language. Phonemes are described by features (See Supplementary Table 2). To generate phonemic representations, we first split each word into its International Phonetic Alphabet (IPA) formatted phonemes using eng-to-ipa[40]. We created a set of 33 binary features for each phoneme using ipapy[41]. The consonant feature categories include voicing, place of articulation, and manner of articulation, and the vowel feature categories include height, backness, voicing, and roundedness. For consistent embedding sizes, we take the union of these features to form our phoneme embeddings. Note that the only shared feature between vowels and consonants is voicing. We concatenate each word's phoneme feature vectors into a larger word-level phonemic embedding. Since words have different numbers of phonemes, we take the maximum number of phonemes in a single word in the podcast (17) and pad the other embeddings with zeros to make the embeddings a consistent size across words (totaling 561 = 17 times 33 since the size of each phoneme embedding is 33). We tried padding just the larger indices of the embeddings as well as padding such that the middle phoneme was always in the middle, and the first and last phonemes were always at the ends. Although we use the latter in our figures, both techniques yielded equivalent results.

Morphemes are the smallest units of meaning in language. Each word is composed of one or more morphemes. For the morphological embeddings, we converted each word in the podcast into a set of morphemes using polyglot[42]. We extracted the set of all morphemes contained in the podcast (1042), and one-hot encoded this set. For each word, we concatenated the one-hot encodings of the morphemes in that word in order. Since words have different numbers of morphemes, we padded this embedding with zeros up to the max embedding length (one-hot encoding size multiplied by the maximum number of morphemes in any word within the podcast, totaling 1042 × 6 = 6252). Thus, the final word-level morphological embedding size is 6252.

For syntactic embeddings, we use the Part of Speech (POS) classifier of Spacy by explosion (See https://spacy.io/models/en), accessible through the trained en_core_web_sm model. Spacy uses the Universal POS TAG set (See https://universaldependencies.org/u/pos) consisting of 17 elements such as Adjectives, Adverbs, and Interjections. Spacy (through OntoNotes 5 dataset[43]) also uses TAGs, a fine-grained (somewhat larger) enumeration of POS, containing 36 refined TAGs (See https://www.ling.upenn.edu/courses/Fall_2003/ling001/penn_treebank_pos.html) such as Coordinating conjunction, Cardinal number, Determiner and so on. The Spacy parser identifies input sentences and related syntactic hierarchy (that is, it creates a parse tree for each sentence), associating with each token its syntactic parent

(i.e., the head of the parse tree of the sentence to which the token belongs) and descendants. We use the POS and TAG of the token itself and of the tokens of the head and the leftmost and rightmost descendants to create an 8-dimensional syntactic embedding for a token. We then one-hot encode this embedding to get an embedding of size 212 (4 tokens times (17 (POS) and 36 (TAG))). If a word is tokenized to more than a single token, we compute the one-hot encoded embeddings for each token and average them to get the embedding for that word.

For semantic embeddings, we again use the trained en_core_web_sm model of Spacy. As documented and detailed in refs. 44 and 45, the model was trained for Named Entity Recognition on the OntoNotes 5 dataset[43], thereby capturing semantic aspects in the underlying embedded representations. The representations are accessible through the (96-dimensional) tensor property of parsed input tokens using the Spacy API. The resulting 96-dimensional embeddings are averaged across tokens for words comprised of more than one token.

These procedures yielded a 4-level hierarchy where each word has a curated embedding for each level. We repeated the encoding-based analyses (as presented in Figs. 2 and 3). Note that before encoding, we applied PCA to reduce the dimensionality of the embeddings to 50. To prevent leakage, we applied PCA separately for each layer and each train-test split, with the projections being learned from the training data only.

The results indicated that LLMs embeddings correlate more with the brain than the curated embedding induced by these psycholinguists' approaches. In addition, while the psychological curated embeddings do correlate with the neural response, the emerging temporal dynamic does not seem to align with the temporal process in the brain ($p > 0.1$; for the results with all words, see Supplementary Fig. 9).

In the implementation we described, the results suggest that the embeddings induced by LLMs better model the temporal dynamic of the brain than the classical symbolic approach. We emphasize that there are many other possible implementations for the phonemic-morphemic-syntactic approach, and we offer this dataset as a benchmark for the community to test them.

### Neural dataset as a benchmark for alternative theories

In this study, we hope to advance the field of computational neuroscience by publishing a comprehensive dataset comprising both lexical stimuli and corresponding neural responses measured across multiple electrodes during the language comprehension experiment. Our dataset includes vectorial representations for each word, allowing for the encoding analysis of linguistic processing in the brain. The inclusion of detailed encoding analyses, notably the standard encoding (illustrated in Fig. 2C), scaled encoding (Fig. 2E), and lag-layer plots (Fig. 2F), serves as a foundational resource for the research community. This publication aims to foster a collaborative environment where diverse theoretical perspectives on language comprehension can be proposed, tested, and refined.

By making this dataset publicly available, we invite the research community to engage in a collective effort to unravel the complexities of the language comprehension process in the brain. This open approach democratizes access to high-quality neural and linguistic data and underscores the importance of transparency and reproducibility in computational neuroscience research. Researchers are encouraged to apply diverse analytical methods and theoretical frameworks to this dataset, potentially leading to insights into the neural substrates of language understanding.

## Discussion

Our results provide evidence for shared computational principles in how LLMs and the human brain process natural language. By leveraging the superior temporal resolution of ECoG in fitting the contextual embeddings of LLMs, we found that the layer-wise transformations learned by LLMs (GPT2-XL and Llama 2) map onto the temporal dynamics of high-level language areas in the brain. The results were reproduced for both predicted and not predicted words, further enhancing their robustness. This finding reveals an important link between how LLMs and the brain process language by demonstrating a correspondence between the temporal neural brain response to spoken language with the hierarchical processing of language in the LLMs.

We also introduce a dataset designed to benchmark models on the temporal hierarchy of comprehension in the brain. While benchmarking performance has been a long-standing tradition in the Machine Learning (ML) and Deep Learning (DL) communities, contributing significantly to the fields' growth, it remains underutilized in computational and cognitive neuroscience. We invite researchers to use this benchmark (and others[46]) to evaluate their computational models of brain comprehension processes. This enables comparisons not only with the results presented here but also with future contributions that we anticipate will enrich this study area.

Utilizing this benchmark, we showcased the enhanced modeling capabilities of LLMs as a potential alternative to the traditional linguistic hierarchy, including phonemic, morphemic, syntactic, and semantic layers. This was evidenced by a superior fit between the LLM embeddings and the temporal brain dynamics observed in human language processing. However, we acknowledge that alternative implementations could yield superior results, perhaps even surpassing the performance of LLMs, thereby significantly contributing to the ongoing research in this field.

In our results, we occasionally observed negative-lag correlations may result from either pre-onset word prediction (reflecting top-down anticipatory processes[1]) or residual encoding of prior contextual features. Because contextual embeddings simultaneously capture both the historical linguistic environment and projections of future linguistic structure, it is challenging to disentangle these processes strictly from correlation time-courses. Future work examining explicit manipulations of predictability (e.g., varying cloze probability) or focusing on temporally fine-grained neural measures might help parse these intertwined mechanisms.

Our study points to implementation differences between the internal sequence of computations in transformer-based LLMs and the human brain. LLMs rely on a transformer architecture, a neural network architecture developed to process hundreds to thousands of words in parallel during training. In other words, transformers are designed to parallelize a task largely computed serially, word by word, in the human brain. While transformer-based LLMs process words sequentially over layers, we found evidence for similar sequential processing in the human brain but over time relative to word onset within a given cortical area. For example, we found that within high-order language areas (such as IFG and TP), a temporal processing sequence corresponded to the layer-wise processing sequence in LLMs. In addition, we demonstrated that this correspondence results from the non-linear transformations across layers in the language model and is not a result of straightforward linear interpolation between the previous and current words (Supplementary Fig. 8).

In our analysis, we observed particularly strong encoding performances in the middle layers of the LLM, especially in the mSTG region (as detailed in Supp Fig. 1 and Supp. Fig. 3). This pattern of heightened correlation aligns with prior findings[3], underscoring the robustness of middle LLM layers in capturing relevant neural activity. The exceptional performance in mSTG may be attributed to its low-level auditory and phonological processing characteristics, which inherently involve reduced temporal uncertainty or noise, resulting in clearer correlations with LLM outputs.

Although our results leverage the hierarchical processing capabilities of LLMs to model temporal aspects of comprehension, they

do not necessarily reflect the intricate dynamics within specific regions like the mSTG. Future studies might benefit from exploring models that incorporate acoustic features, as suggested by research in ref. 47, to provide a more comprehensive model of the auditory pathways and their interaction with cognitive processes. As our results focus on the temporal dynamics of comprehension (at different brain regions), they complement the existing literature that explains human hierarchical spatial responses to speech using deep models trained on textual and auditory domains[3,48–50]. While prior work has suggested possible spatial gradients in timing within the IFG, our results did not reveal significant differences between BA44 and BA45. This may reflect either true homogeneity in timing dynamics across these subregions or limitations in spatial sampling (e.g., absence of BA47 coverage). Further studies with broader coverage may be required to explore these distinctions.

The implementation differences between the brain and language models may suggest that cortical computation within a given language area better aligns with recurrent architectures, where the internal computational sequence is deployed over time rather than over layers. The sequence of temporal processing unfolds over longer timescales as we proceed up the processing hierarchy, from aSTG to IFG and TP. It may be that the layered architecture of LLMs is recapitulated within the local connectivity of a given language area like IFG (rather than across cortical areas). That is, local connectivity within a given cortical area may resemble the layered graph structure of LLMs. To some extent this is supported by recent work that uncovers a deep relation between recurrent-neural-nets (RNNs) and transformers-based models[51], suggesting that from a computational perspective, LMM computations can be modeled by RNNs. It is also possible that long-range connectivity between cortical areas could yield the temporal sequence of processing observed within a single cortical area. Together, these results hint that a deep language model with stacked recurrent networks may better fit the human brain's neural architecture for processing natural language. Interestingly, several attempts have been made to develop neural architectures for language learning and representation, such as universal transformers[52,53] and reservoir computing[54]. Future studies will have to compare how the internal processing of natural language compares between these models and the brain.

Another fundamental difference between LLMs and the human brain is the characteristics of the data used to train these models. Humans do not learn language by reading text. Rather they learn via multi-modal interaction with their social environment. Furthermore, the amount of text used to train these models is equivalent to hundreds (or thousands) of years of human listening. An open question is how LLMs will perform when trained on more human-like input: not textual data but spoken, multimodal, embodied, and immersed in social actions. Interestingly, two recent papers suggest that language models trained on more realistic human-centered data can learn a language like children[55,56]. However, additional research is needed to explore these questions.

The current paper presents evidence that LLMs and the human brain process language in surprisingly similar ways. Despite the clear architectural differences between them, the convergence of their internal computational sequences is noteworthy. While classical psycholinguistic theories propose a rule-based, symbolic system for linguistic processing, LLMs offer a radically different approach–learning language through its statistics by predicting language use in context. This unexpected mapping (layer sequence to temporal dynamics) opens avenues for understanding the brain and for developing neural network architectures that better mimic human language processing.

A critical question for future research is how to align the dynamic sequence of nonlinear neural transformations we observed in language areas and LLMs with the interpretable structure of linguistic processes described in classical rule-based linguistic studies. Influential works, including[57] and[58], have suggested that different layers of LLMs encode increasingly abstract linguistic features. However, recent studies challenge the notion of a neat hierarchical pipeline, wherein each LLM layer serves a fixed linguistic or semantic function. Research by ref. 59 reveals that context length can shift attention toward different linguistic features across layers. Furthermore[60], demonstrate that a layer's function can change depending on the input. These findings cast doubt on the idea that each layer strictly processes one class of linguistic features. With further research, we hope that a better understanding of the internal, context-dependent processes of linguistic information in LLMs and the human brain will emerge.

This study provides compelling evidence of shared internal computations between LLMs and the human brain and suggests a paradigm shift–from a symbolic representation of language to a focus on contextual embeddings and statistical language models. However, recent evidence highlights their limitations in capturing complex sentence structures and replicating empirical human processing effects[61,62]. These findings underscore the importance of systematically evaluating LLMs as cognitive models to delineate both their strengths and expressive boundaries. At the same time, LLMs have demonstrated the ability to approximate traditional linguistic representations[63], suggesting an opportunity to reconcile deep learning approaches with traditional psycholinguistic frameworks. Bridging these approaches through interpretability efforts may offer deeper insights into human cognitive processes and the nature of linguistic representations[64].

## Method
### Data acquisition
Nine patients Five females, M = 31.43 years old, SD = 10.13. The patients listened to the same story stimulus ("So a Monkey and a Horse Walk Into a Bar: Act One, Monkey in the Middle") from beginning to end. We did not include sex or gender as factors in this experiment. We report the effects for individual participants. The audio narrative is 30 min long and consists of 5000 words. All patients volunteered for this study, which was approved by the host institution's Institutional Review Board. All participants had elected to undergo intracranial monitoring for clinical purposes and provided oral and written informed consent before study participation. Language areas were localized to the left hemisphere in all epileptic participants using the WADA test. All epileptic participants were tested for verbal comprehension index (VCI), perceptual organization index (POI), processing speed index (PSI), and Working Memory Index (VMI). See Supplementary Table 3, which summarizes each patient's pathology and neuropsychological scores. In addition, all patients passed the Boston Picture Naming Task and auditory naming task[65,66]. Due to the participants' lack of significant language deficits, our results should generalize outside of our cohort.

Patients were informed that participation in the study was unrelated to their clinical care and that they could withdraw from the study at any point without affecting their medical treatment. After consenting to participate in the experiment, they were told they would hear a 30-min podcast and were asked to listen to it.

For each patient, electrode placement was determined by clinicians based on clinical criteria. One patient consented to have an FDA-approved hybrid clinical research grid implanted, which includes standard clinical electrodes and additional electrodes between clinical contacts. The hybrid grid provides a higher spatial coverage without changing clinical acquisition or grid placement. Across all patients, 1106 electrodes were placed on the left hemisphere and 233 on the right hemisphere. Brain activity was recorded from a total of 1339 intracranially implanted subdural platinum-iridium electrodes embedded in silastic sheets (2.3 mm diameter contacts, Ad-Tech Medical Instrument; for the hybrid grids, 64 standard contacts had a diameter of 2 mm and an additional 64 contacts were 1 mm diameter, PMT corporation, Chanassen, MN). Decisions related to electrode

placement and invasive monitoring duration were determined solely on clinical grounds without reference to this or any other research study. Electrodes were arranged as grid arrays (8 × 8 contacts, 10 or 5 mm center-to-center spacing) or linear strips.

Pre-surgical and post-surgical T1-weighted MRIs were acquired for each patient, and the location of the electrodes relative to the cortical surface was determined from co-registered MRIs or CTs following the procedure described in ref. 67. Co-registered, skull-stripped T1 images were nonlinearly registered to an MNI152 template, and electrode locations were then extracted in Montreal Neurological Institute (MNI) space (projected to the surface) using the co-registered image. All electrode maps are displayed on a surface plot of the template using an Electrode Localization Toolbox for MATLAB. The podcast was played using a laptop, by a research assistant that was blind to the experimental purpose. No additional personnel attended the experiment.

Data were preprocessed using Matlab 2019b and The Fieldtrip toolbox (commit: 56769ab0s). The Python packages used: Scipy v1.11.4, Statsmodels v0.14.1, nltk 3.9.1. Codes for: For embedding extraction: https://github.com/hassonlab/247-pickling/tree/whisper-paper-1For encoding models: https://github.com/hassonlab/247-encoding/tree/whisper-paper-1For plotting: https://github.com/hassonlab/247-plotting/blob/main/scripts/tfspaper_whisper.ipyn

## Preprocessing
66 electrodes from all patients were removed due to faulty recordings. Large spikes in the electrode signals exceeding four quartiles above and below the median were removed, and replacement samples were imputed using cubic interpolation. We then re-referenced the data to account for shared signals across all electrodes using the Common Average Referencing (CAR) or an ICA-based method (based on the participant's noise profile). High-frequency gamma broadband (HFBB) power provided evidence for a high positive correlation between local neural firing rates and high gamma activity. Broadband power was estimated using 6-cycle wavelets to compute the power of the 70–200 Hz band (high-gamma band), excluding 60, 120, and 180 Hz line noise. Power was further smoothed with a Hamming window with a kernel size of 50 ms.

## Linguistic embeddings
To extract contextual embeddings for the stimulus text, we first tokenized the words for compatibility with GPT2-XL. We then ran the GPT2-XL model implemented in HuggingFace (4.3.3)[68] on this tokenized input. To construct the embeddings for a given word, we passed the set of up to 1023 preceding words (the context) along with the current word as input to the model. The embedding we extract is the output generated for the previous word. This means that the current word is not used to generate its own embedding and its context only includes previous words. We constrain the model in this way because our human participants do not have access to the words in the podcast before they are said during natural language comprehension.

GPT2-XL is structured as a set of blocks, each containing a self-attention sub-block and a subsequent feedforward sub-block. The output of a given block is the summation of the feedforward output and the self-attention output through a residual connection. This output is also known as a hidden state of GPT2-XL. We consider this hidden state to be the contextual embedding for the block that precedes it. For convenience, we refer to the blocks as layers; that is, the hidden state at the output of block 3 is referred to as the contextual embedding for layer 3. To generate the contextual embeddings for each layer, we store each layer's hidden state for each word in the input text. The HuggingFace implementation of GPT2-XL automatically stores these hidden states when a forward pass of the model is conducted. Different models have different numbers of layers and embeddings of different dimensionality. The model used herein, GPT2-XL, has 48 layers, and the embeddings at each layer are 1600-dimensional vectors. For a sample of text containing 101 words, we would generate an embedding for each word at every layer, excluding the first word as it has no prior context. This results in 48 1600-dimensional embeddings per word and with 100 words; 48 * 100 = 4800 total 1600-long embedding vectors. Note that the context length would increase from 1 to 100 in this example as we proceed through the text.

## Dimensionality reduction
Before fitting the encoding models, we reduce the dimensionality of the embeddings from each layer separately by applying principal component analysis (PCA) and retaining the first 50 components. This procedure effectively focuses our subsequent analysis on the 50 orthogonal dimensions in the embedding space that account for the most variance in the stimulus. We do not compute PCA on the entire set of embeddings (layers x words) as that would result in mixing information between layers. However, our main results computed PCA on the concatenation of train and test folds for a given layer and predictability condition (predicted or not predicted words). To ensure our results are not impacted by leakage between train and test data, we fit PCA only on the training folds and used this projection matrix on the test fold (for each test fold, we learned a separate PCA projection matrix). The results were reproduced and are reported in Supplementary Fig. 10.

## Encoding models
Linear encoding models were estimated at each lag (-2000 ms to 2000 ms in 25-ms increments) relative to word onset (0 ms) to predict the brain activity for each word from the corresponding contextual embedding. Before fitting the encoding model, we smoothed the signal using a rolling 200-ms window (i.e., for each lag, the model learns to predict the average signal + − 100 ms around the lag). For the effect of the window size on the results, see Supplementary Fig. 11. We used a tenfold cross-validation procedure, ensuring that for each cross-validation fold, the model was estimated from a subset of training words and evaluated on a non-overlapping subset of held-out test words: the words and the corresponding brain activity was split into a training set (90% of the words) for model estimation and a test set (10% of the words) for model evaluation. Encoding models were estimated separately for each electrode and each lag relative to word onset. We used ordinary least squares (OLS) multiple linear regression for each cross-validation fold to estimate a weight vector (50 coefficients for the 50 principal components) based on the training words. We then used those weights to predict the neural responses for the test words. We repeated this for the remaining test folds to obtain results for all words. We evaluated model performance by computing the correlation between the predicted brain activity at test time and the actual brain activity across words (for a given lag); we then averaged these correlations across electrodes within specific ROIs. This procedure was performed for all the layers in GPT2-XL to generate an encoding for each layer.

## Predicted and not predicted words
After generating embeddings for all words in the podcast transcript, we split the embeddings into two subsets: words that the model could predict and words that the model could not. A word was considered predicted if the model assigned that word the highest probability of occurring next among all possible words. We refer to this subset of embeddings as top 1 predicted or just predicted (1709 words out of 4744 = 36%). To reduce the stringency of top 1 prediction, we also created subsets of top 5 predicted (2936 words out of 4744 = 62%) and top 5 not predicted words where the criterion for predictability was that the probability assigned by the model to predicted words must be among the highest five probabilities assigned by the model. The top 5 Not predicted words, which we also refer to as simply Not predicted words, were not in that subset. As outlined above, we then trained linear encoding models on the top 1 predicted and top 5 not predicted embedding sets.

## Statistical significance

To establish the significance of the bars in Fig. 2B we conducted a bootstrapping analysis for each lag. Given the values of the electrodes in a specific layer and a specific ROI, we sampled the values of the max correlations with replacement ($10^4$ samples, including values for all electrodes). We computed the mean for each sample and generated a distribution (consisting of $10^4$ points). We then compared the actual mean for the lag-ROI pair to estimate how significant it is given the generated distributions. The results can be seen in Supplementary Fig. 2. The '*' indicates a two-tailed significance of $p < 0.01$.

## Electrode selection

We used a nonparametric statistical procedure with correction for multiple comparisons[69] to identify significant electrodes. We randomized each electrode's signal phase at each iteration by sampling from a uniform distribution. This disconnected the relationship between the words and the brain signal while preserving the autocorrelation in the signal. We then performed the encoding procedure using GloVe embeddings for each electrode (for all lags). We chose to focus on electrodes demonstrating significant encoding performance with GloVe embeddings to avoid concerns of circular reasoning–specifically, selecting electrodes based on their significance with GPT-2 layers and then showing that they align with the dynamics of those layers. We repeated this process 5000 times. After each iteration, the encoding model's maximal value across all lags was retained for each electrode. We then took the maximum value for each permutation across electrodes. This resulted in a distribution of 5000 values, which was used to determine the significance for all electrodes. For each electrode, a $p$ value was computed as the percentile of the non-permuted encoding model's maximum value across all lags from the null distribution of 5000 maximum values. Performing a significance test using this randomization procedure evaluates the null hypothesis that there is no systematic relationship between the brain signal and the corresponding word embedding. This procedure yielded a $p$-value per electrode, corrected for the number of models tested across all lags within an electrode. To further correct for multiple comparisons across all electrodes, we used a false-discovery rate (FDR). Electrodes with $q$-values less than 0.01 are considered significant. This procedure identified 160 electrodes in the left hemisphere's early auditory, motor cortex, and language areas. We used subsets of this list corresponding to the IFG ($n = 46$), TP ($n = 6$), aSTG ($n = 13$), and mSTG ($n = 28$).

## Controlling for the best-performing layer

We divided our results into all combinations of { predicted, not predicted, all words} x {mSTG, TP, aSTG, IFG} and found the layer with maximum encoding correlation (highest peak in the un-normalized encoding plot) for each combination (max-layer see Supplementary Table 4). For each layer, we projected its embeddings onto the embeddings for the max-layer using the dot product (separate projection per word). We then subtracted these projections from the layer's embeddings to get a set of embeddings for that layer that are orthogonal to their projections onto the max layer. We also project the max-layer from itself to ensure the information was removed properly (seen in black in Supplementary Fig. 6). We then ran encoding on these sets of embeddings as normal (Supplementary Fig. 6 for the IFG result). Our finding of a temporal ordering of layer-peak correlations in the IFG is preserved.

## Interpolation significance test

To show that the contextual embeddings generated through the layered computations in GPT2 are significantly different from those generated through a simple linear interpolation between the input layer (previous word) and output layer (current word), we linearly interpolated ~$10^3$ embeddings between the first and last contextual embeddings of GPT2-XL. We then re-ran our lag layer analysis for 10,000 iterations (for each ROI and predictability condition), except instead of using the 48 layers of GPT2-XL, we used the first and last layers' embeddings, and 46 intermediate layers sampled without replacement from the set of linear interpolations and then sorted. We constructed a distribution of correlations between layer index (the sampled layers were sorted and assigned indices 2–47) and the corresponding lags that maximize the encodings for each layer. We then computed the $p$-value of our true correlation for that combination of ROI and word classification condition using this distribution. The results are in Supplementary Fig. 8.

## Not predicted words

The temporal correspondence described in the main text was observed for words the model accurately predicted; We conducted the same layer-wise encoding analyses in the same ROIs for not predicted words–i.e., words for which the probability assigned to the word was not among the top-5 highest probabilities assigned by the model ($N = 1808$). See the 'not predicted' column in Supplementary Figs. 2, 3, 4 for these results. We still see evidence, albeit slightly weaker, for layer-based encoding sequences in the IFG ($r = 0.81$, $p < 10e–11$) and TP ($r = 0.57$, $p < 10e–4$), but not aSTG ($r = 0.09$, $p > 0.55$) or mSTG ($r = −0.10$, $p > 0.48$). Similar results were obtained with Spearman correlation (mSTG $r = −0.10$, $p > 0.48$; aSTG $r = 0.02$, $p > 0.9$; IFG $r = 0.8$, $p < 10e–11$; TP $r = 0.72$, $p < 10e–8$), demonstrating that the effect is robust to outliers. We conducted permutation tests that yielded the following $p$ values: $p > 0.24$ (mSTG), $p > 0.27$ (aSTG), and $p < 10e–5$ (TP, IFG). While we observed a sequence of temporal transitions across layers in language areas, we did not observe such transformations in mSTG. The lack of temporal sequence in mSTG may be since it is sensitive to speech-related phonemic information rather than word-level linguistic analysis[70–73].

We noticed a difference in the times of peak correlation across layers between the predicted and not predicted words in the IFG (See Supplementary Fig. 3). We ran a paired t-test to compare the average of lags (over the electrodes in an ROI) that yield the maximal correlations (i.e., peak encoding performance) across predicted and unpredicted words for each layer. The paired $t$ test indicated that the shift of the lag of peak encoding (at the ROI level) was significant for 9 out of the 12 first layers (corrected for multiple comparisons, see Supplementary Table 5, $q < 0.01$).

## Compute

This work employed two computationally intensive processes: the generation of GPT2-XL embeddings and the training of our encoding models. We allocated 1 GPU and 4 CPUs, 192GB of memory, and 6 h for the former process. To train encoding models for all electrodes and lags, we allocated 4 CPUs and 30GB of memory. This process took only 3 min but had to be repeated for all layers and predictability conditions. To reproduce the core analyses in the main body of our paper, with the 48 layers and predicted words, would have taken around 2.5 h. The computational resources used were on a shared cluster at the institution where the computational aspects of this work were conducted.

## Reporting summary

Further information on research design is available in the Nature Portfolio Reporting Summary linked to this article.

# Data availability

Source data are provided with this paper. The language model evaluation dataset used in this study is available in the public GitHub repository under accession code GPTology. The dataset is openly accessible and does not require special permission. The raw and processed data used in this study are included in the repository. Full reference: Vader, D. & Reichart, R. (2023). GPTology: A unified

framework for analyzing the inner workings of GPT models. GitHub repository: https://github.com/DVader96/GPTology. Source data are provided with this paper.

## Code availability

The code used to perform the analyses and generate the figures presented in the main body of this work is available at the following public GitHub repository: https://github.com/DVader96/GPTology. The code for the brain plots is described at[74].

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

## Acknowledgements

We thank our funders: NIH NINDS R01NS109367, NIH NINDS R01NS115929, DP1HD091948, R01DC022534.

## Author contributions

A.G.-devised the project, performed experimental design and data analysis, wrote the paper; E.H.- devised the project,performed experimental design and data analysis, wrote the paper; M.S.-devised the project, performed data analysis; S.A.N.-critically revised the article, wrote the paper; B.A.-performed data collection; Z.Z.-performed experimental design, data analysis; A.G.D.- data analysis; H.G.- data analysis; A.Fe.-critically revised the article; W.D.- devised the project; S.D.- devised the project; P.D.-devised the project; D.F.- devised the project; M.B.- devised the project; A.H.- devised the project; Y.M.-devised the project; O.D.- devised the project; N.S.-designed control analyses; A.F.l.-performed experimental design; O.L.- devised the project; R.R.- critically revised the article; U.H.-devised the project,performed experimental design and data analysis,wrote the paper.

## Competing interests

The authors declare no competing interests.
