## [Transparent Peer Review file · Nature Communications]

Temporal Structure of Natural Language Processing in the Human Brain Corresponds to Layered Hierarchy of Large Language Models

Corresponding Author: Dr Ariel Goldstein

Version 0:

Reviewer comments:

Reviewer #1

(Remarks to the Author)

Summary

In this study, the authors investigate linear relationships between temporal dynamics of intracranially recorded brain signals (ECoG) during 30-minute story listening and contextual vector representations of words in the story ("word embeddings") extracted from internal layers of two language models (LMs) at various processing depths (shallow to deep layers). They report that layer hierarchy in LMs maps onto temporal dynamics in the brain: word representations from deep LM layers, better predict brain responses later relative to word onsets and vice versa. That is, that there exist some correspondence between the processing steps in an LLM and temporal dynamics of electrophysiological signals.

Over all, this is already a strong and polished manuscript and I have no problem seeing this published. The work is based on previous published work, but certainly has elements of novelty. The authors performed a range of control analyses and reporting is for the most part thorough and clear. They also acknowledge limitations where appropriate.

I do have some remarks that I would love to see authors address in a revision before I can recommend this for publication. Primarily, I'd like to challenge the authors on some of their interpretations and certain aspects of evidence reported in the manuscript that seem to speak against the account they offer.

I think the authors should not have difficulty responding to these points. I hope my comments are clear and that the authors find my feedback useful.

Main points

1. LMs as an alternative to "traditional psycholinguistics". While I appreciate your effort in pushing forward the use of data-driven tools such as deep neural networks and LMs, the "traditional psycholinguistic" and deep learning frameworks are not necessarily at odds, as I am sure you are aware. For example, continuous vector representations (embeddings) can well approximate/learn discrete representations in a continuous space (or discrete representation can be converted into continuous vectors as you demonstrate). Given this, I think you should at least acknowledge that a) LMs may in principle recover "traditional linguistic representations" (e.g. as in Manning, 2020, that you cite) and b) that there is also evidence that LMs fail to model empirical human processing effects (e.g. Huang et al, 2024, <https://doi.org/10.1016/j.jml.2024.104510>, Lakretz et al, 2022, <https://aclanthology.org/2022.coling-1.285> for complex sentence constructions). I think this can be just a sentence or two in the Introduction or Discussion, but it would be useful to signal these empirical facts for balance and as contrast to your main narrative.

2. Spatial patterns of encoding performance. It looks like the middle LM layers have by far the strongest encoding performance in mSTG (peak rho of ~0.3 in Figures 4. and Supp Fig 6). Which at the same time is also the area that shows no lag-layer relationship. What do you make of that result? Wouldn't your hierarchical processing account predict that earlier LM layers (as opposed to middle/late) would perform best in mSTG? It seems to me that this data point is possibly not in line with your main interpretation of temporal processing hierarchy in LLMs. For completeness, it would be good to address the

high encoding performance of signals in mSTH.

3. Interpretation of high encoding performance at negative lags (Figure 3). One of your fundamental "shared principles" of language comprehension are next-word prediction (2) and error-correction (3). In your previous work, you interpreted high (non-zero) encoding at negative lags as evidence of pre-onset word prediction and high encoding at later lags as evidence of post-onset prediction error. Here, the TP and aSTG results show high encoding at negative lags for early LLM layers and you interpret early encoding peaks for early LLM layers as evidence of (feed-forward?) temporal processing hierarchy. So, is encoding performance at negative lags evidence of (top-down?) prediction or bottom-up/feed-forward processing? Aren't these two accounts (predictive vs. temporal processing hierarchy) of encoding dynamics slightly at odds? If so, how do you (propose to) reconcile that?

4. Regressing out middle layers (Supp Fig 9). Can you clarify the motivation for this analysis? You state "In order to emphasize the relationship between the unique contributions of the intermediate representations induced by the layers and the dynamic of language comprehension in the brain", but I find it unclear. Is the concern that most of the late layer results are driven by middle layers (due to feed-forward LLM architecture)? I think it would be helpful to rephrase the motivation somewhat, unless I am missing something. Also, note that you refer to the wrong figure in the main text (this result, I believe, is reported in in Supp Fig 9, not Figure 8 that you refer to).

5. Dimensionality reduction of psycholinguistic embeddings. You mention that you retained 50 PCA dimensions of your 4-level "psycholinguistic embedding". It's not easy to find in the current manuscript what was the final dimension of the "psycholinguistic embedding" for each token? Is it 561 (phonemes) + 1024 (morphemes) + 17 (syntactic) + 96 (semantic) = 1698?

6. Syntactic embeddings, averaging across subtokens. You state when you describe syntactic embeddings that "If a word is tokenized to more than a single token, the resulting embedding used for a word is the average of its tokens' embeddings." But isn't syntactic embedding vector categorical, derived from POS tags (possibly 1-hot encoded)? What was being averaged exactly for syntactic embeddings?

7. Fig Supp 11, sampling pseudo interpolations. You generated linearly interpolated intermediate layers 10,000 times to generate a null distribution that operationalized linear embedding-transformation hypothesis. Specifically, you state "We generated 46 intermediate 'pseudo-layers' by linearly interpolating between the first and last layers. We repeated this process 10,000 times...". In Supp Figure 11, what does the dashed line represent? You state that it represents "correlation achieved in the even-space case", but aren't there 10,000 of such correlations if you created 10,000 pseudo-embeddings? Unless I'm missing something, it would be useful to clarify this procedure.

8. Benchmarking. I like that you're releasing code and data! But you state that benchmarking "remains underutilized in computational and cognitive neuroscience." This might be true overall, but I do think you should at least acknowledge and cite the Brain Score initiative (<https://www.brain-score.org/>) which has a language component too.

Minor points/typos

1. You have malformed in-text citations in several places in the manuscript, e.g.: "hierarchical representations of phonology, morphology, syntax, and semantics Fromkin et al. (2014); Yule (2022)."" (use `\citep{}`).

2. Psycholinguistic Embeddings. Figure 4. Color-coding here is a bit unfortunate as several colors (blue, red, green) map onto brain ROIs and embedding levels. I would at least consider also adding the legend for embedding levels (phonemic, morphemic etc) next to the upper row of Figure 4, next to the line plots where the colors for the 4 embedding levels are actually used. It was hard to parse the figure. Just a suggestion.

Reviewer #2

(Remarks to the Author)

In this study, Goldstein et al present an analysis of the brain response to the listening of natural speech, and show that several cortical areas typically associated with speech processing generate a sequence of neural responses that aligns with the sequence of representations generated across the layers of modern language models.

Overall, this work is technically sound and provides an interesting new piece of evidence highlighting the functional alignment between Language Models and the human brain.

I have some comments but believe they can all be addressed by the authors.

1. Major

1.1 Contributions

Several studies have shown anatomical and/or temporal alignments between the human brain and deep neural networks trained on language.

In particular,

- Li et al (Nature Neuro 2023) showed with a similar encoding analysis that the ECoG responses to speech tend to be best modeled by the first layers of several speech models, namely wav2vec 2.0 and Hubert, whereas later responses are best modeled by deeper layers.

- Millet et al (Neurips, 2022) as well as Vaidya et al (PMLR, 2022) showed with fMRI-encoding that there exists a systematic alignment between AI speech models and the hierarchy of the language network.

- Caucheteux & King (Nature Communications Biology 2022) showed with both fMRI and MEG that word embedding (i.e. the first layer of a language model) and the deeper layer of language models (1) account for the response in different brain regions and (2) these encoding scores peak at different moments. Specifically, during this reading task, many areas are first accounted for in the first layer of the language model, and later on becomes better predicted by its contextual layers.

I do believe the present study does complement those results in a valuable way: the type of brain recordings, the language modality, the exact deep learning model and/or the analysis vary substantially across these studies. However, I think it is important to better explain how the present study complements and/or differs from previous results.

1.2 Numbers of patients

It is not clear whether the effects are consistent across participants. For example, it could be that only 1/9 patients had electrodes in the infero-frontal gyrus, in which case it would be difficult to generalize the present finding to the general population.

I would recommend the author to add the number of patients included in the electrode cluster below each region of interest in each figure.

1.3 Stats

Many/most stats are applied across either time samples or electrodes. This is problematic as, in each of these two approaches, the samples are not independent from one another.

For example, in Figure 2F, the statistics is indicated as $r=0.85$, $p<1e-13$. However, it is the correlation between the layer index and the lag. Consequently, the degrees of freedom, and thus p-value, directly depend on the number of layers. For example, a much higher p-value would be obtained from e.g. GPT-2-small, even if it equally predicted brain activity, and showed a similar temporal alignment.

In sum, the present p-values do not reflect the confidence of observing the reported effect should we repeat this experiment (i.e. with a new participant).

I am conscious that statistics across participants are often omitted in intracranial research, often because the number of participants is low. However, I would strongly recommend the author to provide the across-subjects statistics whenever possible, so as to provide a better estimate of the confidence we may have in these observations.

The same argument goes for statistics across electrodes. The authors indicate “Significance was assessed using bootstrap resampling across electrodes”. However, electrodes are not independent of one another. The resulting p-value is thus misleading.

2. Minor

2.1 Interpretation approximation

The authors indicate that they “observed a temporal sequence in our encoding results where earlier layers yield peak encoding performance relative to word onset, and later layers yield peak encoding performance later in time. This finding suggests that the transformation sequence across layers in LLMs maps onto a temporal accumulation of information in high-level language areas”

The authors implicitly link the notion of representation delay and temporal accumulation. However, the latter concept refers to a specific notion in computational neuroscience (e.g. Shadlen et al), where one can demonstrate that a neural system accumulates evidence (typically over time).

I am not convinced that the present results show a temporal accumulation. Rather, it shows a similar *transformation* of representations between the AI model and the brain.

2.2 PCA

The authors used PCA ($n_{\text{components}}=50$) over the model embeddings before fitting the regression.

This step is not highly motivated, as it could remove components which do not account for a large variance in the embedding, but may nevertheless be (potentially highly) predictable of the brain data.

It is unclear how this preprocessing decision impacts the analysis.

2.3 Encoding model

The author seems to use a simple ordinary least square regression for their encoding model. This is unusual, as most studies typically used a regularized regression (typically ridge regression)

2.4 Cross val

The authors indicate: "To evaluate the linear model, we used the 50-dimensional weight vector estimated from the 9 training set folds to predict the neural signals in the held-out test set fold."

I believe the authors estimated the linear model from each training set and evaluated on the held out test set. However, the present wording is ambiguous, and could suggest that the weights resulted from the aggregation of the training sets, which would be invalid.

2.5 Scoring metrics

The authors indicate: "We evaluated the model's performance by computing the correlation between the predictions and true values for all words."

This suggests that the correlation was not performed with each split (which is preferable, as a non identically distributed splitting would lead to distribution shift across splits, which themselves could lead to different predictions, and thus biases in the evaluation metrics.)

2.6 Data

The authors should clarify in the main text whether they show results from the broadband (bipolar constructs?) or gamma responses.

2.7 Predictability

The results on word predictability are only present in the supplementary analyses but not discussed.

2.8 Discussion references

The authors indicate: "This finding reveals an important link between how LLMs and the brain process language: conversion of discrete input into multidimensional (vectorial) embeddings, which are further transformed via a sequence of nonlinear transformations to match the context-based statistical properties of natural language (Manning et al., 2020)."

The reference used to support the discussion point could be improved. The Manning paper primarily focuses on the vectorial representations of syntactic trees, which is not the topic of the present study which investigates either LLM representations (without distinguishing syntax or semantics) or syntactic classes like part-of-speech.

The paper by Caucheteux et al ICML 2021 'Disentangling syntax and semantics in the brain with deep networks' could be a relevant reference to discuss given the authors' consideration of syntax and semantics.

Jean-Rémi KING

Reviewer #3

(Remarks to the Author)

Review of A. Goldstein et al., 2024 : Temporal Structure of Natural Language Processing in the Human Brain corresponds to Layered Hierarchy of Large Language Models

Summary:

This article examines ECoG data from nine epileptic patients as they listen to speech. The authors explore the relationship between the layers of two large language models, GPT-2 XL and Llama, and the patients' brain activity. They achieve this by identifying language-related regions of interest (ROIs) and modeling each electrode's activity based on the LLMs' embeddings. The findings reveal that in the inferior frontal gyrus (IFG), as the model layer progresses, the peak of correlation occurs later in time relative to the onset of a word, indicating that deeper layers are predictive at later stages. Additionally,

the layer with the strongest correlation is a middle layer. This pattern generalizes to other language regions, where they observe that the higher a region is in the language hierarchy, the greater the variation in peak timing within the region. The authors regard this dataset as a valuable benchmark for testing current and future theories and make it available to the community.

Evaluation:

This article offers interesting insights into the relationship between the depth of LLM layers and the temporal dynamics of language processing mechanisms. However, several additional analyses are needed to draw interesting conclusions. The study lacks results on the information represented by the LLM layers and does not fully utilize ECoG, as it does not include fine-grained spatial analysis. As it stands, it provides very limited insights into similarities in the brain and the LLMs and does not advance our understanding of language processing mechanisms in either the brain or LLMs.

General comments:

Spatial resolution of ECoG data

According to Hagoort in *Broca's Brain and Binding: A New Framework* (2005), the function of the Inferior Frontal Gyrus (IFG) in language processing is not uniform. There is a gradient from posterior to anterior regions of the IFG, with each region representing different aspects of language, such as phonology, syntax, and semantics. It would be highly valuable if the authors conducted a spatial analysis to examine the timing of the peak correlation with the LLM's embeddings. Moreover, it is a bit disappointing not to see any spatial analysis from a method known to its spatial resolution.

Selected language ROIs

The selection of the language ROIs, as considered by the authors, warrants further discussion. Notably, the pSTS is missing from their analysis, despite its well-established role in language processing. Did the authors exclude this region due to the lack of electrodes placed there? It would be essential to have a map showing the distribution of electrodes across different participants.

Selection of relevant electrodes based on GLOVE

It seems quite surprising to use electrodes demonstrating significant encoding performance for GLOVE static embeddings when the objective is to explore correlations with contextual LLMs. By doing so, the authors might overlook some electrodes in the dmPFC, precuneus, and angular gyrus, which are known to play a role in high-level language interpretation and for which contextual models have proven to be better than GLOVE.

Which type of information are the different layers representing?

The authors do not investigate the nature of the information represented by the various layers, nor do they analyze correlations between the embeddings and phonemic, morphemic, syntactic, or semantic traits. Although they reference the work of N. Sahin et al., *Science*, 2009, they do not draw inspiration from it, despite its findings on a processing hierarchy within Broca's area.

Considering only a subpart of the trials : the accurately predicted words

- 1) The authors chose to analyse only the words that were accurately predicted by the LLM (30% of Top-1), which means picking the trials in which their model works. Even if this practice is spread in the community, it needs to be justified.
- 2) Moreover, I disagree with the terms 'Predictable' and 'Unpredictable' to describe the words that were predicted versus those that were not. These adjectives seem to imply inherent qualities of the words themselves, whereas this categorization is based on the LLM's performance. I would recommend using 'Predicted' and 'Not Predicted' instead.
- 3) When comparing the symbolic model to the LLMs, the authors should include the entire dataset rather than only the 'Predicted' conditions, as this approach would otherwise unfairly favour their model.

Critiques on the discussion section :

- 1) The discussion lacks consistency and many statements unjustified such as :

This finding reveals an important link between how LLMs and the brain process language: conversion of discrete input into multidimensional (vectorial) embeddings, which are further transformed via a sequence of nonlinear transformations to match the context-based statistical properties of natural language.

The authors have only shown that, in language related areas, early layers show an early correlation with the brain signals and deep layers show a later correlation.

- 2) Furthermore, it may seem surprising to conclude "Finally, this paper provides strong evidence that LLMs and the brain process language similarly." after a paragraph on how RNNs may better account for how the human brain processes language as they represent information sequentially, unlike LLMs.

An interesting reference for this discussion may be M. Oren et al., *Transformers are Multi-State RNNs*, <https://arxiv.org/html/2401.06104v1> 2024 where the authors discuss the fact that transformers may be an asymptotic approximation of what RNNs do.

3) No discussion of the comparison between the results for GPT2-XL and Llama is done. Are there differences? If not, why? How similar are these two architectures?

4) The dataset highlighted by the authors has already been the subject of a previous study by the same authors (nature communication, 2022)

Minor points:

In part 3, you refer to figure 8. It's actually figure 9.

Figure 3: From the presented results, it is not clear that there is any significant correlation between electrodes in mSTG and the model. Could you also plot the correlation before rescaling? The fact that all layers exhibit a peak decoding exactly at the same time, before the word onset, is not discussed and seems a bit strange. Is there anticipation? What type of information is presented at that exact moment?

Figure 10 : The y labels are missing for the four panels of the figure.
There is a problem with the x label and the numbers could be printed in a larger font.

Figure 9 : What is represented in thick black lines?

On Figure 9 : precedes the onset, what is the correlation? surprising that peak is before the word onset. Can you discuss in the main text such anticipation effect?

Version 1:

Reviewer comments:

Reviewer #2

(Remarks to the Author)

The authors adequately addressed my original comments.

Reviewer #3

(Remarks to the Author)

While I appreciate the authors' efforts in addressing some of my previous comments, I still have significant concerns regarding the following points.

As I mentioned before, the lack of spatial analysis investigating the timing of peak correlations across different subregions of the IFG is a major weakness. Your statement that you "did not observe any unique dynamics in the timing" without further exploration or explanation is insufficient. I strongly recommend conducting more in-depth spatial analyses, such as finer-grained parcellation or statistical tests for differences in timing across subregions. Alternatively, provide a compelling explanation for why you didn't find the expected spatial dynamics, considering potential limitations of your data or methodology.

I believe the study could gain significant value by exploring the interpretability of the LLM layers and their relationship to language processing in the human brain. Unfortunately, the authors relegate this important aspect to future work.

REVIEWER COMMENTS

Reviewer #1 (Remarks to the Author):

Summary

In this study, the authors investigate linear relationships between temporal dynamics of intracranially recorded brain signals (ECoG) during 30-minute story listening and contextual vector representations of words in the story ("word embeddings") extracted from internal layers of two language models (LMs) at various processing depths (shallow to deep layers). They report that layer hierarchy in LMs maps onto temporal dynamics in the brain: word representations from deep LM layers, better predict brain responses later relative to word onsets and vice versa. That is, that there exist some correspondence between the processing steps in an LLM and temporal dynamics of electrophysiological signals.

Over all, this is already a strong and polished manuscript and I have no problem seeing this published. The work is based on previous published work, but certainly has elements of novelty. The authors performed a range of control analyses and reporting is for the most part thorough and clear. They also acknowledge limitations where appropriate.

I do have some remarks that I would love to see authors address in a revision before I can recommend this for publication. Primarily, I'd like to challenge the authors on some of their interpretations and certain aspects of evidence reported in the manuscript that seem to speak against the account they offer.

I think the authors should not have difficulty responding to these points. I hope my comments are clear and that the authors find my feedback useful.

Main points

1. *LMs as an alternative to "traditional psycholinguistics". While I appreciate your effort in pushing forward the use of data-driven tools such as deep neural networks and LMs, the "traditional psycholinguistic" and deep learning frameworks are not necessarily at odds, as I am sure you are aware. For example, continuous vector representations (embeddings) can well approximate/learn discrete representations in a continuous space (or discrete representation can be converted into continuous vectors as you demonstrate). Given this, I think you should at least acknowledge that a) LMs may in principle recover "traditional linguistic representations" (e.g. as in Manning, 2020, that you cite) and b) that there is also evidence that LMs fail to model empirical human processing effects (e.g. Huang et al, 2024, <https://doi.org/10.1016/j.jml.2024.104510>, Lakretz et al, 2022, <https://aclanthology.org/2022.coling-1.285> for complex sentence constructions). I think this can be just a sentence or two in the Introduction or*

Discussion, but it would be useful to signal these empirical facts for balance and as contrast to your main narrative.

We appreciate the reviewer's insightful comments regarding the relationship between "traditional psycholinguistic" frameworks and deep learning approaches. We agree that these frameworks are not inherently at odds and recognize the potential for large language models (LLMs) to approximate traditional linguistic representations through continuous vector spaces, as demonstrated by Manning (2020).

At the same time, we acknowledge the limitations of LLMs in modeling certain human language processing phenomena, as highlighted in recent empirical studies. For instance, LLMs struggle with complex sentence constructions and fail to capture some human processing effects (Huang et al., 2024; Lakretz et al., 2022).

To address this, we have added the following paragraph to the Discussion section:

This study provides compelling evidence of shared internal computations between LLMs and the human brain and suggests a paradigm shift—from a symbolic representation of language to a focus on contextual embeddings and statistical language models. However, recent evidence highlights their limitations in capturing complex sentence structures and replicating empirical human processing effects (Huang et al., 2024; Lakretz et al., 2022). These findings underscore the importance of systematically evaluating LLMs as cognitive models to delineate both their strengths and expressive boundaries. At the same time, LLMs have demonstrated the ability to approximate traditional linguistic representations (Manning, 2020), suggesting an opportunity to reconcile deep learning approaches with traditional psycholinguistic frameworks. Bridging these approaches through interpretability efforts may offer deeper insights into human cognitive processes and the nature of linguistic representations.

- 2. Spatial patterns of encoding performance. It looks like the middle LM layers have by far the strongest encoding performance in mSTG (peak rho of ~0.3 in Figures 4. and Supp Fig 6). Which at the same time is also the area that shows no lag-layer relationship. What do you make of that result? Wouldn't your hierarchical processing account predict that earlier LM layers (as opposed to middle/late) would perform best in mSTG? It seems to me that this data point is possibly not in line with your main interpretation of temporal processing hierarchy in LLMs. For completeness, it would be good to address the high encoding performance of signals in mSTH.*

Thank you for your insightful observation regarding the spatial patterns of encoding performance, particularly in the middle layers of the LLM as they relate to mSTG. This pattern is consistently observed across all regions of interest (ROIs), with the middle layers of the LM correlating more strongly compared to others, a phenomenon also reported by others in the field (Caucheteux et al 2022). The relatively higher correlation in mSTG, as opposed to other regions, may reflect its "lower-level" processing characteristics, which could result in reduced temporal uncertainty or noise, thus enhancing signal clarity and encoding strength. It is possible

the the neural response is more locked to word onset in the mSTG thus the higher correlations in general to other ROIs.

Regarding the hierarchical processing interpretation, our current work posits that different layers of the LLM model various temporal processes associated with language comprehension, without necessarily mirroring the dynamic processes within the mSTG itself. This suggests that while our LLM framework provides valuable insights into broader cognitive processes, specific neural dynamics, especially those related to more acoustic features in regions like mSTG, might be better modeled by approaches that focus more directly on these features, such as those discussed in Li et al (2023).

We added the following paragraph to the paper:

In our analysis, we observed particularly strong encoding performances in the middle layers of the LLM, especially in the mSTG region (as detailed in Fig. 4 and Supp. Fig. 6). This pattern of heightened correlation aligns with prior findings (Caucheteux et al., 2022), underscoring the robustness of middle LLM layers in capturing relevant neural activity. The exceptional performance in mSTG may be attributed to its low-level auditory and phonological processing characteristics, which inherently involve reduced temporal uncertainty or noise, resulting in clearer correlations with LLM outputs.

While our results leverage the hierarchical processing capabilities of LLMs to model temporal aspects of comprehension, they do not necessarily reflect the intricate dynamics within specific regions like the mSTG. Future studies might benefit from exploring models that incorporate acoustic features, as suggested by research in (Li et al 2023), to provide a more comprehensive model of the auditory pathways and their interaction with cognitive processes. This integration could bridge the gap between our current understanding and the complex reality of brain function, enhancing the interpretability and applicability of LLMs in cognitive neuroscience.

- 3. Interpretation of high encoding performance at negative lags (Figure 3). One of your fundamental "shared principles" of language comprehension are next-word prediction (2) and error-correction (3). In you previous work, you interpreted high (non-zero) encoding at negative lags as evidence of pre-onset word prediction and high encoding at later lags as evidence of post-onset prediction error. Here, the TP and aSTG results show high encoding at negative lags for early LLM layers and you interpret early encoding peaks for early LLM layers as evidence of (feed-forward?) temporal processing hierarchy. So, is encoding performance at negative lags evidence of (top-down?) prediction or bottom-up/feed-forward processing? Aren't these two accounts (predictive vs. temporal processing hierarchy) of encoding dynamics slightly at odds? If so, how do you (propose to) reconcile that?*

We appreciate the reviewer for highlighting this important point, which we did not address in the original manuscript or reconcile with our previous work. To clarify: in Goldstein et al. (2022), we examined pre-onset predictions using static embeddings (GloVe and arbitrary) to establish how the brain anticipates upcoming words (see Fig. 4 in Goldstein et al., 2022). In that study, we

intentionally avoided to make this interpretation based on contextual embeddings because it was unclear whether the observed correlations reflected next-word prediction or information related to the preceding context. In this current paper, as we utilize contextual embeddings, pre-onset correlations could potentially reflect either predictive processes or the integration of prior contextual information. We have added further clarification in the revised manuscript to address this distinction.

We added the following to the paper:

Negative-lag correlations may result from either pre-onset word prediction reflecting top-down anticipatory processes, as previously described in Goldstein et al. 2022 or residual encoding of prior contextual features. Because contextual embeddings simultaneously capture both the historical linguistic environment and projections of future linguistic structure, it is challenging to disentangle these processes strictly from correlation time-courses. Future work examining explicit manipulations of predictability (e.g., varying cloze probability) or focusing on temporally fine-grained neural measures might help parse these intertwined mechanisms

4. *Regressing out middle layers (Supp Fig 9). Can you clarify the motivation for this analysis? You state "In order to emphasize the relationship between the unique contributions of the intermediate representations induced by the layers and the dynamic of language comprehension in the brain", but I find it unclear. Is the concern that most of the late layer results are driven by middle layers (due to feed-forward LLM architecture)? I think it would be helpful to rephrase the motivation somewhat, unless I am missing something.*

We appreciate the reviewer's comments regarding the motivation for regressing the contributions of middle layers. Our observation prompted this analysis that embeddings from intermediate layers consistently show superior performance across different time lags, suggesting they might capture essential brain activity related to language comprehension. The primary concern was that the observed correlations between brain activity and the embeddings from layers other than the intermediate, 'optimal' layers might be due to their similarity to this optimal layer, rather than reflecting unique information encoded by each layer. To address this, we conducted an analysis where we regressed the influence of the optimal layer embeddings. This approach allowed us to demonstrate that the significant correlations with other layers are not merely due to their resemblance to the optimal layers but are attributable to unique information that these layers independently encode.

We added the following to the paper:

In response to our observation that embeddings from intermediate layers consistently outperform those from other layers at various time lags, we conducted a regression analysis to investigate the source of the correlations between brain activity and layer embeddings. We hypothesized that the effectiveness of non-optimal layers might primarily derive from their similarities to the 'optimal' intermediate layers. By regressing the influence of embeddings from these intermediate layers, our analysis revealed that the significant correlations with other layers are due to unique information that they independently encode. This finding underscores each

layer's embeddings' independent and distinct contributions to our understanding of neural dynamics in language comprehension.

5. Also, note that you refer to the wrong figure in the main text (this result, I believe, is reported in Supp Fig 9, not Figure 8 that you refer to).

We thank the reviewer for the careful reading. We fixed the mistake.

6. Dimensionality reduction of psycholinguistic embeddings. You mention that you retained 50 PCA dimensions of your 4-level "psycholinguistic embedding". It's not easy to find in the current manuscript what was the final dimension of the "psycholinguistic embedding" for each token? Is it 561 (phonemes) + 1024 (morphemes) + 17 (syntactic) + 96 (semantic) = 1698?

We thank the reviewer for pointing out this omission from our manuscript. We separately compute embeddings for phonemes (size = 561), morphemes (6252), syntax (212), and semantics (96). For each, we reduce the dimensionality to 50 using PCA. We then run encoding separately on each linguistic "layer".

For morphemes, there are 1042 total morphemes in the podcast so we have one-hot encodings of size 1042 for each morpheme. To construct word-level morphological embeddings, we concatenate the one-hot encodings of the morphemes in each word. The word with the most morphemes has 6 morphemes, so its size is 6252. We pad all other word-level morphological embeddings with zeros to make the word-level morphological embeddings all of size 6252.

We modified the sections on morphological and syntactic embeddings to make the final embedding size clearer. The edits are marked in red in pages 9-10.

7. Syntactic embeddings, averaging across subtokens. You state when you describe syntactic embeddings that "If a word is tokenized to more than a single token, the resulting embedding used for a word is the average of its tokens' embeddings." But isn't syntactic embedding vector categorical, derived from POS tags (possibly 1-hot encoded)? What was being averaged exactly for syntactic embeddings?

We thank the reviewer for pointing out the lack of clarity on our part. The averaging was performed on the one hot encodings of the tokens for syntax, not on the tokens themselves. We clarified this in the paper.

8. *Fig Supp 11, sampling pseudo interpolations. You generated linearly interpolated intermediate layers 10,000 times to generate a null distribution that operationalized linear embedding-transformation hypothesis. Specifically, you state "We generated 46 intermediate 'pseudo-layers' by linearly interpolating between the first and last layers. We repeated this process 10,000 times...". In Supp Figure 11, what does the dashed line represent? You state that it represents "correlation achieved in the even-space case",*

but aren't there 10,000 of such correlations if you created 10,000 pseudo-embeddings? Unless I'm missing something, it would be useful to clarify this procedure.

We thank the reviewer for the opportunity to clarify this analysis. Conceptually, each embedding represents a point in a high-dimensional space. We focused on two such points: one from the first hidden layer and one from the last hidden layer. We then created a “line” between these two points and sampled 46 positions along it. In doing so, we generated 10,000 sets of 46 “pseudo-layer” embeddings—these sets were *not* necessarily evenly spaced along that line.

In addition, we also created a single, *evenly spaced* set of 46 positions between the first and last layers. The dashed line in Supplemental Figure 11 represents the correlation value achieved by this *evenly spaced* set of pseudo-layer embeddings, whereas the colored distributions represent the correlations generated by the 10,000 random (i.e., not evenly spaced) sets. We have updated the figure legend to reflect this distinction and to clarify that the dashed line corresponds specifically to the evenly spaced case, while the gray distribution (or histogram) corresponds to the 10,000 pseudo-layer sets.

We added the following to the paper:

The lag-layer slope for the evenly spaced pseudo-layers (dashed line) is compared to the actual lag-layer analysis (solid line) for predictable words. The p-value is derived from panel B.

9. *Benchmarking. I like that you're releasing code and data! But you state that benchmarking "remains underutilized in computational and cognitive neuroscience." This might be true overall, but I do think you should at least acknowledge and cite the Brain Score initiative (<https://www.brain-score.org/>) which has a language component too.*

We thank the reviewer for bringing to our attention this benchmark, which we were not familiar with. We rephrased the relevant part of the discussion to acknowledge this source.

We added the following to the paper:

We invite researchers to use this benchmark (and others, such as <https://www.brain-score.org/>) to evaluate their computational models of brain comprehension processes.

Minor points/typos

1. You have malformed in-text citations in several places in the manuscript, e.g.: "hierarchical representations of phonology, morphology, syntax, and semantics Fromkin et al. (2014); Yule (2022)."" (use `\citep{}`).

Thank you for the comment. We fixed it.

2. Psycholinguistic Embeddings. Figure 4. Color-coding here is a bit unfortunate as several colors (blue, red, green) map onto brain ROIs and embedding levels. I would at least

consider also adding the legend for embedding levels (phonemic, morphemic etc) next to the upper row of Figure 4, next to the line plots where the colors for the 4 embedding levels are actually used. It was hard to parse the figure. Just a suggestion.

Thank you very much for this suggestions. We adopted it.

Reviewer #2 (Remarks to the Author):

In this study, Goldstein et al present an analysis of the brain response to the listening of natural speech, and show that several cortical areas typically associated with speech processing generate a sequence of neural responses that aligns with the sequence of representations generated across the layers of modern language models.

Overall, this work is technically sound and provides an interesting new piece of evidence highlighting the functional alignment between Language Models and the human brain.

I have some comments but believe they can all be addressed by the authors.

1. *Contributions* Several studies have shown anatomical and/or temporal alignments between the human brain and deep neural networks trained on language. In particular,
 - Li et al (Nature Neuro 2023) showed with a similar encoding analysis that the ECoG responses to speech tend to be best modeled by the first layers of several speech models, namely wav2vec 2.0 and Hubert, whereas later responses are best modeled by deeper layers.

We thank the reviewer for this insightful comment. We agree that Li et al. (Nature Neuro, 2023) is highly relevant to our claims, as it also highlights the relationship between the hierarchical structure of deep learning models and human neural processes during comprehension. However, it is crucial to note that the models used by Li et al. are fundamentally different, as they are trained on acoustic signals, which may influence their application to earlier neural processes in regions such as the AN, IC, HG, and STG. Their study primarily addresses lower-level linguistic processes, including syllable and phonetic processing, with an emphasis more on neuroanatomical aspects rather than the temporal dynamics within brain regions. This distinction underscores the need to consider the specific characteristics and training of the models when interpreting their relevance to neural processing in different contexts.

To acknowledge their relevance, we added to the paper:

In our analysis, we observed particularly strong encoding performances in the middle layers of the LLM, especially in the mSTG region (as detailed in Fig. 4 and Supp. Fig. 6). This pattern of heightened correlation aligns with prior findings (Caucheteux et al., 2022), underscoring the robustness of middle LLM layers in capturing relevant neural activity. The exceptional performance in mSTG may be attributed to its low-level auditory and phonological processing characteristics, which inherently involve reduced temporal uncertainty or noise, resulting in clearer correlations with LLM outputs. While our hierarchical processing framework leverages LLMs to model temporal aspects of comprehension, it does not fully capture the intricate dynamics within regions like mSTG.

While our results leverage the hierarchical processing capabilities of LLMs to model temporal aspects of comprehension, they do not necessarily reflect the intricate dynamics within specific regions like the mSTG. Future studies might benefit from exploring models that incorporate acoustic features, as suggested by research in (Li et al 2023), to provide a more comprehensive model of the auditory pathways and their interaction with cognitive processes. This integration could bridge the gap between our current understanding and the complex reality of brain function, enhancing the interpretability and applicability of LLMs in cognitive neuroscience.

2. - Millet et al (Neurips, 2022) as well as Vaidya et al (PMLR, 2022) showed with fMRI-encoding that there exists a systematic alignment between AI speech models and the hierarchy of the language network.

- Caucheteux & King (Nature Communications Biology 2022) showed with both fMRI and MEG that word embedding (i.e. the first layer of a language model) and the deeper layer of language models (1) account for the response in different brain regions and (2) these encoding scores peak at different moments. Specifically, during this reading task, many areas are first accounted for in the first layer of the language model, and later on becomes better predicted by its contextual layers.

I do believe the present study does complement those results in a valuable way: the type of brain recordings, the language modality, the exact deep learning model and/or the analysis vary substantially across these studies. However, I think it is important to better explain how the present study complements and/or differs from previous results.

We thank the reviewer for pointing out these papers. We absolutely agree that our results complement these results; thus we added them to our paper as follows:

As our results focus on the temporal dynamics of comprehension (at different brain regions), they complement the existing literature that explains human hierarchical spatial responses to speech using deep models trained on textual and auditory domains (Michler et al 2024, Caucheteux & King 2022, Millet et al 2022; Vaidya et al 2022).

3. 1.2 Numbers of patients

It is not clear whether the effects are consistent across participants. For example, it could be that only 1/9 patients had electrodes in the infero-frontal gyrus, in which case it would be difficult to generalize the present finding to the general population. I would recommend the author to add the number of patients included in the electrode cluster below each region of interest in each figure.

We thank the reviewer for this fully justified comment. To address this concern, we have added a supplementary table that specifies the number of electrodes for each patient across brain areas (Table Supp. 6). This table provides a detailed breakdown of the electrode distribution, including the statistical significance of the correlations (noted by p values).

The table highlights that the effects are observed across multiple participants and regions, ensuring that the findings are not driven by a single individual (in fact all patients with electrodes in the IFG showed the effect). While there is some variability in the distribution of electrodes across participants, the consistent statistical significance across regions underscores the robustness of our results. We added to the result section:

for the results per participant per ROI please see Supp. Table 6

patient	mSTG	aSTG	IFG	TP
1	11**(negative)	6***	15***	0
2	0	0	2***	0
3	10*	6***	10***	4***
4	4***	0	1***	0

5	0	0	18***	1***(negative)
6	2	0	0	0
7	1**	1***	0	1**

*p<.05 **p<.01 ***p<.001

Supp. Table 6. Distribution of electrodes per participant over the different ROIs. The significance addresses the correlation between the lag of the peak correlation and the index layer.

We note that in two cases, negative correlations were reached. For patient 1 in the mSTG we can see that it is a result of a mild trend induced by spurious correlations.

And for patient 5 in the TP we can see the time frame (-150 - -200 ms before word onset) is not relevant to comprehension:

4. 1.3 Stats

Many/most stats are applied across either time samples or electrodes. This is problematic as, in each of these two approaches, the samples are not independent from one another.

For example, in Figure 2F, the statistics is indicated as $r=0.85$, $p<1e-13$. However, it is the correlation between the layer index and the lag. Consequently, the degrees of freedom, and thus p -value, directly depend on the number of layers. For example, a much higher p -value would be obtained from e.g. GPT-2-small, even if it equally predicted brain activity, and showed a similar temporal alignment.

In sum, the present p -values do not reflect the confidence of observing the reported effect should we repeat this experiment (i.e. with a new participant).

I am conscious that statistics across participants are often omitted in intracranial research, often because the number of participants is low. However, I would strongly recommend the author to provide the across-subjects statistics whenever possible, so as to provide a better estimate of the confidence we may have in these observations.

The same argument goes for statistics across electrodes. The authors indicate “Significance was assessed using bootstrap resampling across electrodes”. However, electrodes are not independent of one another. The resulting p -value is thus misleading.

We thank the reviewer for their insightful comments. We fully acknowledge the importance of addressing the independence of samples in our statistical analyses, particularly concerning correlations across time samples and electrodes, as highlighted in the manuscript.

To address the concerns raised, we have revised our statistical approach by implementing a Linear Mixed Model (LMM) that accounts for the non-independence of both electrodes and subjects. This model includes both fixed and random effects, where **layer order** and **region of interest (ROI)** are treated as fixed effects, while **electrodes** and **participants** are considered random effects. This statistical framework allows us to generalize our findings more confidently across patients and electrodes.

Specifically, the updated analysis examines the relationship between **layer order** and **latency** across ROIs, with the model structured as follows:

Model: lag ~ 1 + layer + ROI + (1 + layer | electrode) + (1 + layer | participant)

This approach accounts for variability across both participants and electrodes while assessing the fixed effects of layer order and region of interest. The results of this analysis show that all fixed effects are significant ($p < .001$), indicating that the influence of layer order on lag (of highest correlation) is robust and generalizable across different electrodes and patients.

Specifically, we updated our analysis as follows:

We ran a linear mixed model for the relationship between layer order and latency across ROIs, structured to generalize across electrodes and participants (Model: lag ~ 1 + layer + ROI + (1 + layer | electrode) + (1+layer | participant)). All fixed effects in this model were significant ($p < .001$), indicating that the influence of layer order on latency is robust, extending across different electrodes and patients.

5. Minor

2.1 Interpretation approximation

The authors indicate that they “observed a temporal sequence in our encoding results where earlier layers yield peak encoding performance relative to word onset, and later layers yield peak encoding performance later in time. This finding suggests that the transformation sequence across layers in LLMs maps onto a temporal accumulation of information in high-level language areas”

The authors implicitly link the notion of representation delay and temporal accumulation. However, the latter concept refers to a specific notion in computational neuroscience (e.g. Shadlen et al), where one can demonstrate that a neural system accumulates evidence (typically over time).

*I am not convinced that the present results show a temporal accumulation. Rather, it shows a similar *transformation* of representations between the AI model and the brain.*

We appreciate the reviewer's careful analysis and thoughtful critique of our terminology. We acknowledge that the term "accumulation" may inadvertently imply a process of evidence accumulation over time, which is a specific concept in computational neuroscience and is not what our data directly demonstrate.

We have revised the terminology in our manuscript to clarify our findings and better align our language with the observed phenomena. Instead of "accumulation," we now use "dynamic" to describe the representation changes across the language model's layers. This term more accurately reflects our observations that each layer in the language model correlated to a progression in the temporal dynamics of information processing rather than suggesting a cumulative build-up of evidence.

The revised sentence states:

This finding suggests that the transformation sequence across layers in LLMs maps onto a temporal dynamic of information process in high-level language areas.

6. 2.2 PCA

The authors used PCA ($n_{\text{components}}=50$) over the model embeddings before fitting the regression. This step is not highly motivated, as it could remove components which do not account for a large variance in the embedding, but may nevertheless be (potentially highly) predictable of the brain data. It is unclear how this preprocessing decision impacts the analysis.

7. 2.3 Encoding model

The author seems to use a simple ordinary least square regression for their encoding model. This is unusual, as most studies typically used a regularized regression (typically ridge regression)

We appreciate the reviewer's observation regarding the prevalent use of ridge regression in similar studies. In our research, we adhered to established encoding and preprocessing protocols outlined in two prior publications: Goldstein et al., *Nature Communications* (2024), and Goldstein et al., *Nature Neuroscience* (2022). These references serve as benchmarks for our methodological approach and underpin the analytical choices made in this study. Specifically, we utilized principal component analysis (PCA) for dimensionality reduction, given the dataset's size—1,709 correctly predicted and 1,800 incorrectly predicted words, each represented by GPT-2 XL embeddings with 1,600 dimensions. We believe that our approach, informed by prior studies and tailored to the demands of our dataset, provides a robust framework for analysis, ensuring our findings are both reliable and comparable to established research in the field.

However, in response to the reviewer's request, we reproduced the results using the ridge regression procedure and confirmed that our findings remain consistent:

7. 2.4 Cross val

The authors indicate: “To evaluate the linear model, we used the 50-dimensional weight vector estimated from the 9 training set folds to predict the neural signals in the held-out test set fold.”

I believe the authors estimated the linear model from each training set and evaluated on the held out test set. However, the present wording is ambiguous, and could suggest that the weights resulted from the aggregation of the training sets, which would be invalid.

We thank the reviewer for this comment. Indeed, we estimated the linear model from each training set and evaluated it on the held-out test set. We edited the encoding model description as follows:

We estimated the linear model from each 9-training folds and evaluated it on the held-out test fold.

8. 2.5 Scoring metrics

The authors indicate: “We evaluated the model’s performance by computing the correlation between the predictions and true values for all words.”

This suggests that the correlation was not performed with each split (which is preferable, as a non identically distributed splitting would lead to distribution shift across splits, which themselves could lead to different predictions, and thus biases in the evaluation metrics.)

We thank the reviewer for this insightful comment. We would like to emphasize that the analyses in our paper focus primarily on the temporal dynamics, which are determined by the encoding correlation (metric evaluation). Additionally, we followed the same procedures reported in our previous work to ensure consistency and to avoid biasing the results toward

expected outcomes. However, in response to the reviewer's suggestion, we reanalyzed the data following their proposed approach and successfully replicated the main results.

9. 2.6 Data

The authors should clarify in the main text whether they show results from the broadband (bipolar constructs?) or gamma responses.

We thank the reviewer for this comment. It made us realize that this information is not addressed in the body of the paper, thus we edited the following:

We collected electrocorticography (ECoG) data from 9 epilepsy patients (7 of them had electrodes in the pre-defined ROIs) while they listened to a 30-minute audio podcast (“Monkey in the Middle”, NPR 2017), and preprocessed the neural data to reflect the high-gamma band signal (see A.2).

In the preprocessing section, we state that:

High-frequency gamma broadband (HFBB) power provided evidence for a high positive correlation between local neural firing rates and high gamma activity. Broadband power was estimated using 6-cycle wavelets to compute the power of the 70-200 Hz band (high-gamma band), excluding 60, 120, and 180 Hz line noise. Power was further smoothed with a Hamming window with a kernel size of 50 ms.

10. 2.7 Predictability

The results on word predictability are only present in the supplementary analyses but not discussed.

We thank the reviewer for the comment. We edited section 2.1:

To further separate correct and incorrect predictions and to match the statistical power across the analyses, we defined incorrect predictions as cases where all top-5 next-word predictions were incorrect. In other words, the correct word was not in the 5 most probable next words set as determined by GPT2-XL. There were 1808 of these words. In Supp. Figs. 6 & 7, we ran our analysis for correct, incorrect, and all words.

We also added to the discussion:

The results were reproduced for both predicted and not predicted words, further enhancing their robustness.

11. 2.8 Discussion references

The authors indicate: "This finding reveals an important link between how LLMs and the brain process language: conversion of discrete input into multidimensional (vectorial) embeddings, which are further transformed via a sequence of nonlinear transformations to match the context-based statistical properties of natural language (Manning et al., 2020)." . The reference used to support the discussion point could be improved. The Manning paper primarily focuses on the vectorial representations of syntactic trees, which is not the topic of the present study which investigates either LLM representations (without distinguishing syntax or semantics) or syntactic classes like part-of-speech. The paper by Caucheteux et al ICML 2021 'Disentangling syntax and semantics in the brain with deep networks' could be a relevant reference to discuss given the authors' consideration of syntax and semantics.

We thank the reviewer for this comment and included the offered citation in the sentence.

Jean-Rémi KING

Reviewer #3 (Remarks to the Author):

Review of A. Goldstein et al., 2024 : Temporal Structure of Natural Language Processing in the Human Brain corresponds to Layered Hierarchy of Large Language Models

Summary:

This article examines ECoG data from nine epileptic patients as they listen to speech. The authors explore the relationship between the layers of two large language models, GPT-2 XL and Llama, and the patients' brain activity. They achieve this by identifying language-related regions of interest (ROIs) and modeling each electrode's activity based on the LLMs' embeddings. The findings reveal that in the inferior frontal gyrus (IFG), as the model layer progresses, the peak of correlation occurs later in time relative to the onset of a word, indicating that deeper layers are predictive at later stages. Additionally, the layer with the strongest correlation is a middle layer. This pattern generalizes to other language regions, where they observe that the higher a region is in the language hierarchy, the greater the variation in peak timing within the region. The authors regard this dataset as a valuable benchmark for testing current and future theories and make it available to the community.

Evaluation:

This article offers interesting insights into the relationship between the depth of LLM layers and the temporal dynamics of language processing mechanisms. However, several additional analyses are needed to draw interesting conclusions. The study lacks results on the information

represented by the LLM layers and does not fully utilize ECoG, as it does not include fine-grained spatial analysis. As it stands, it provides very limited insights into similarities in the brain and the LLMs and does not advance our understanding of language processing mechanisms in either the brain or LLMs.

General comments:

1. *Spatial resolution of ECoG data*

According to Hagoort in Broca's Brain and Binding: A New Framework (2005), the function of the Inferior Frontal Gyrus (IFG) in language processing is not uniform. There is a gradient from posterior to anterior regions of the IFG, with each region representing different aspects of language, such as phonology, syntax, and semantics. It would be highly valuable if the authors conducted a spatial analysis to examine the timing of the peak correlation with the LLM's embeddings. Moreover, it is a bit disappointing not to see any spatial analysis from a method known to its spatial resolution.

We thank the reviewer for this insightful comment. Following the suggestion, we conducted a spatial analysis focusing on electrodes in BA44 and BA45, which, along with BA47, constitute the IFG. These regions are known to play distinct roles in language processing, as highlighted by Hagoort's framework describing a gradient from posterior to anterior IFG regions corresponding to phonology, syntax, and semantics. In our analysis, the IFG demonstrated consistently high lag-layer correlations ($r > 0.74$). However, we did not observe any unique dynamics in the timing of the peak correlations across these regions (BA44/45). We appreciate the emphasis on the spatial resolution of ECoG data and will consider further exploration of finer spatial dynamics in future work.

2. Selected language ROIs

The selection of the language ROIs, as considered by the authors, warrants further discussion. Notably, the pSTS is missing from their analysis, despite its well-established role in language processing. Did the authors exclude this region due to the lack of electrodes placed there? It would be essential to have a map showing the distribution of electrodes across different participants.

We thank the reviewer for this comment. The reviewer is correct that we did not include the pSTS, even though it is documented to be highly relevant for language processing. Because our ECoG electrodes were primarily placed on the gyri, none were located in the pSTS, which typically lies in a sulcus. Indeed, ECoG electrodes are usually placed on the cortex for clinical monitoring, often excluding deeper sulcal areas. We acknowledge this limitation in the paper:

Due to the electrode placements, we did not have electrodes in the pSTS.

3. *Selection of relevant electrodes based on GLOVE*

It seems quite surprising to use electrodes demonstrating significant encoding performance for GLOVE static embeddings when the objective is to explore correlations with contextual LLMs. By doing so, the authors might overlook some electrodes in the dmPFC, precuneus, and angular gyrus, which are known to play a role in high-level language interpretation and for which contextual models have proven to be better than GLOVE.

We thank the reviewer for this thoughtful comment. We initially chose to focus on electrodes demonstrating significant encoding performance with GloVe embeddings to avoid concerns of circular reasoning—specifically, selecting electrodes based on their significance with GPT-2 layers and then showing that they align with the dynamics of those layers. We agree that this point needs to be explicit thus we added to the paper:

We chose to focus on electrodes demonstrating significant encoding performance with GloVe embeddings to avoid concerns of circular reasoning—specifically, selecting electrodes based on their significance with GPT-2 layers and then showing that they align with the dynamics of those layers.

Additionally, we note that when we used the electrodes significant for GPT-2's last layers the results were reproduced for the mSTG, aSTG and TP but did not reveal additional significant electrodes in the dmPFC. As such, this region is not included in the current analysis. We added to the paper the following:

4. *Which type of information are the different layers representing?*

The authors do not investigate the nature of the information represented by the various layers, nor do they analyze correlations between the embeddings and phonemic, morphemic, syntactic, or semantic traits. Although they reference the work of N. Sahin et al., Science, 2009, they do not draw inspiration from it, despite its findings on a processing hierarchy within Broca's area.

We thank the reviewer for this comment. We mention Sahin et al 2009 as it is one of the foundational papers in ECoG for modeling neural response for speech. We also agree that while we offer an alternative view to the classic view of phonemic, morphemic, syntactic, semantic hierarchy, the relation between these categories and the embedding is highly relevant and should be addressed. Thus we added the following to the discussion:

At the same time, LLMs have demonstrated the ability to approximate traditional linguistic representations (Manning, 2020), suggesting an opportunity to reconcile deep learning approaches with traditional psycholinguistic frameworks. Bridging these approaches through interpretability efforts may offer deeper insights into human cognitive processes and the nature of linguistic representations (Caucheteux et al 2021).

5. *Considering only a subpart of the trials : the accurately predicted words.*

The authors chose to analyse only the words that were accurately predicted by the LLM (30% of Top-1), which means picking the trials in which their model works. Even if this practice is spread in the community, it needs to be justified.

We thank the reviewer for this comment. We agree that analysis of the incorrect prediction is called for. We noticed that we did not refer to them. So we make sure to refer to the results indicating the pattern is replicated. The main results are replicated for the IFG and TP as can be seen in Supp. Fig. 5. We reproduce the full encoding and scaled encoding in Supp. Fig. 6&7

We added the following to the paper:

To further separate correct and incorrect predictions and to match the statistical power across the analyses, we defined incorrect predictions as cases where all top-5 next-word predictions were incorrect. In other words, the correct word was not in the 5 most probable next words set as determined by GPT2-XL. There were 1808 of these words. In Supp. Figs. 6 & 7, we ran our analysis for correct, incorrect and all words.

Figure 5: Temporal hierarchy along the ventral language stream for **not predicted** words. (Top) The location of electrodes on the brain is color-coded by roi, with blue, black, red, and green corresponding to TP, IFG, aSTG, and mSTG, respectively. (Middle) Scaled encoding performance for these ROIs. (Bottom) Scatter plot of the lag that yields peak encoding performance for each layer.

6. Moreover, I disagree with the terms 'Predictable' and 'Unpredictable' to describe the words that were predicted versus those that were not. These adjectives seem to imply

inherent qualities of the words themselves, whereas this categorization is based on the LLM's performance. I would recommend using 'Predicted' and 'Not Predicted' instead.

We thank the reviewer for this comment and we change the terminology accordingly.

7. *When comparing the symbolic model to the LLMs, the authors should include the entire dataset rather than only the 'Predicted' conditions, as this approach would otherwise unfairly favour their model.*

We thank the reviewer for this comment. We added the analyses of all words.

In addition, while the psychological curated embeddings do correlate with the neural response, the emerging temporal dynamic does not seem to align with the temporal process in the brain ($p > 0.1$; Fig. 4B; for the results of all words see Supp Fig. 14).

8. *Critiques on the discussion section :*

The discussion lacks consistency and many statements unjustified such as: This finding reveals an important link between how LLMs and the brain process language: conversion of discrete input into multidimensional (vectorial) embeddings, which are further transformed via a sequence of nonlinear transformations to match the context-based statistical properties of natural language.

We thank the reviewer for this comment. We edited to reflect the findings themselves:

This finding reveals an important link between how LLMs and the brain process language by mapping the temporal neural brain response with the hierarchical process in the LLMs.

9. *The authors have only shown that, in language related areas, early layers show an early correlation with the brain signals and deep layers show a later correlation. Furthermore, it may seem surprising to conclude “Finally, this paper provides strong evidence that LLMs and the brain process language similarly.” after a paragraph on how RNNs may better account for how the human brain processes language as they represent information sequentially, unlike LLMs. An interesting reference for this discussion may be M. Oren et al., Transformers are Multi-State RNNs, <https://arxiv.org/html/2401.06104v1> 2024 where the authors discuss the fact that transformers may be an asymptotic approximation of what RNNs do.*

We thank the reviewer for this comment and for the citation as it offers a way to bridge a gap we imply in the paper.

Our results provide strong evidence for a relationship between the neural response to free speech and the hierarchical processing of language by LLMs. Specifically, we show that early layers of LLMs align with early neural responses, while deeper layers correspond to later neural activity, suggesting a shared temporal structure for integrating linguistic context. However, as we discuss in the manuscript, the brain has fundamental properties—such as its recurrent processing of information—that, from an architectural viewpoint, might be better modeled by RNNs (or other state-space models). While LLMs capture the *sequence of contextual representations* observed in brain dynamics, they may not fully account for these additional traits. The citation brought by the reviewer offers a bridge that connects RNN with transformer based models. Thus we added the following to the discussion in the paper:

The implementation differences between the brain and language models may suggest that cortical computation within a given language area better aligns with recurrent architectures, where the internal computational sequence is deployed over time rather than over layers. The sequence of temporal processing unfolds over longer timescales as we proceed up the processing hierarchy, from aSTG to IFG and TP. It may be that the layered architecture of LLMs is recapitulated within the local connectivity of a given language area like IFG (rather than across cortical areas). That is, local connectivity within a given cortical area may resemble the layered graph structure of LLMs. To some extent this is supported by recent work that uncovers a deep relation between recurrent-neural-nets (RNNs) and transformers-based models (Oren et al. 2024). Suggesting that from a computational perspective LMMs computations can be modeled by RNNs.

3) *No discussion of the comparison between the results for GPT2-XL and Llama is done. Are there differences? If not, why? How similar are these two architectures?*

We thank the reviewer for this comment. Llama-2 has the same architecture as GTP2-XL in the sense that it is composed of stacked decoders, and is trained on next-word prediction. The idea is to show a replication of the results on a newer model than GPT-2XL in order to be able to

make claims about LLMs (which practically are stacked decoder trained using next word prediction) and not just GPT-2XL. We include this explanation in the revised manuscript.

..both models are trained with the objective of next-word prediction and consist of stacked decoder architectures.

Regarding the comparison between the results for GPT-2 XL and LLaMA-2, we did not include a detailed discussion in the current version, and we acknowledge this oversight. In our analysis, we observed no significant differences between the two models in terms of the measured correlations and dynamics. This similarity can be attributed to the fact that both models share very similar architectures and training objectives. Our goal was to replicate the findings using a more recent model, thereby extending the generalizability of our claims to LLMs in general, given that most modern LLMs follow the stacked decoder architecture and next-word prediction paradigm. We will ensure this reasoning and any relevant observations are clearly discussed in the revised manuscript.

10. 4) The dataset highlighted by the authors has already been the subject of a previous study by the same authors (nature communication, 2022)

We thank the reviewer for this comment and for pointing out the prior use of the dataset in our *Nature Communications* (2024) study. The reviewer is correct that the dataset has previously been explored; however, it was only made available for collaboration at that time. In the current work, we are releasing the dataset in its entirety, fully preprocessed and systematically organized as a benchmark. This benchmark is specifically designed to test claims regarding the hierarchical organization of models and the temporal dynamics of language processing. By making the dataset openly accessible and standardized, we aim to facilitate further research and comparison across studies in this domain.

Minor points:

1. In part 3, you refer to figure 8. It's actually figure 9.

We thank the reviewer. We fixed it.

Figure 3: From the presented results, it is not clear that there is any significant correlation between electrodes in mSTG and the model. Could you also plot the correlation before rescaling?

We thank the reviewer for this comment. The unscaled encodings are attached in Sup. Fig. 6. The encoding correlations vary between 0.2 to 0.3.

The fact that all layers exhibit a peak decoding exactly at the same time, before the word onset, is not discussed and seems a bit strange. Is there anticipation?

We appreciate the reviewer for highlighting this important point. In this study, we utilize contextual embeddings, which means that pre-onset correlations could potentially reflect either predictive processes or the integration of prior contextual information. We have added further clarification in the revised manuscript to address this distinction. In Goldstein et al. (2022), we specifically examined pre-onset predictions using static embeddings (e.g., GloVe and arbitrary embeddings) to investigate how the brain anticipates upcoming words (see Fig. 4 in Goldstein et al., 2022). In that study, we intentionally avoided using contextual embeddings to distinguish whether the observed correlations reflected next-word prediction or were influenced by preceding contextual information. This differentiation remains an important area for exploration, and we are grateful for the opportunity to further elaborate on it in our current work.

What type of information is presented at that exact moment? The type of information that is coded and induces the correlation merits further research and is currently beyond the scope of this study.

Figure 9 : What is represented in thick black lines?

We thank the reviewer. We added the following explanation to the paper:

(B) The encoding for each layer after controlling for the variance explained by the optimal layer. The encoding by the optimal layer, shown in black, represents the residual signal after projecting out the variance that can be explained by the same layer (close to 0).

On Figure 9 : precedes the onset, what is the correlation? surprising that peak is before the word onset. Can you discuss in the main text such anticipation effect?

We thank the reviewer for this comment. Please see the above answers.

REVIEWER COMMENTS

Reviewer #2 (Remarks to the Author):

The authors adequately addressed my original comments.

We thank Reviewer 2 for acknowledging we addressed their original points.

Reviewer #3 (Remarks to the Author):

(1) While I appreciate the authors' efforts in addressing some of my previous comments, I still have significant concerns regarding the following points.

We thank Reviewer 3 for acknowledging we addressed some of the previous comments. .

(2) As I mentioned before, the lack of spatial analysis investigating the timing of peak correlations across different subregions of the IFG is a major weakness. Your statement that you "did not observe any unique dynamics in the timing" without further exploration or explanation is insufficient. I strongly recommend conducting more in-depth spatial analyses, such as finer-grained parcellation or statistical tests for differences in timing across subregions. Alternatively, provide a compelling explanation for why you didn't find the expected spatial dynamics, considering potential limitations of your data or methodology.

We thank the reviewer for the opportunity to expand on our previous comment. Following the reviewer's suggestion to do statistical tests for differences in timing across subregions, we compared the lags of the peaks (per layer) between the BA44 ($M=155$ ms, $SD=95$) and BA45 ($M=152$ ms, $SD=72$). Both are part of IFG (we do not have electrodes in BA47). We used a paired sample t-test, as the lags are paired per layer. The difference did not turn significant ($t(47)=0.26, p=.79$). We also used a Bayesian sampled t-test, which resulted in a Bayes factor of 6. Since the Bayes factor is below 10, the Bayesian sample does not provide strong evidence for a difference between the legs of the two groups.

We added the following:

In the Results section:

To test whether timing dynamics differed across subregions of the IFG, we compared the peak lags (per layer) between BA44 ($M = 155$ ms, $SD = 95$) and BA45 ($M = 152$ ms, $SD = 72$). A paired-sample t-test revealed no significant difference ($t(47) = 0.26$, $p = .79$). A Bayesian paired-sample t-test yielded a Bayes factor of 6. Since the Bayes factor is below 10, the Bayesian sample does not provide strong evidence for a difference between the legs of the two groups. These findings suggest that timing dynamics were consistent across the sampled IFG subregions.

In the Discussion section:

While prior work has suggested possible spatial gradients in timing within the IFG, our results did not reveal significant differences between BA44 and BA45. This may reflect either true homogeneity in timing dynamics across these subregions or limitations in spatial sampling (e.g., absence of BA47 coverage). Further studies with broader coverage may be required to explore these distinctions.

- (3) I believe the study could gain significant value by exploring the interpretability of the LLM layers and their relationship to language processing in the human brain. Unfortunately, the authors relegate this important aspect to future work.

Thank you for your valuable comment. We fully agree that the connection between the interpretability of LLM layers and language processing in the human brain is both fascinating and crucial. This is a key focus of an ongoing research initiative in our lab. However, due to the complexity of the current paper, this topic falls outside the scope of this study. Therefore, to address the reviewer's request, we have included a new paragraph in the discussion that highlights the significance of LLM's interpretability for future research.

A critical question for future research is how to align the dynamic sequence of nonlinear neural transformations we observed in language areas and LLMs with the interpretable structure of linguistic processes described in classical rule-based linguistic studies. Influential works, including Belinkov et al. (2017) and Tenney et al. (2019), have suggested that different layers of LLMs encode increasingly abstract linguistic features. However, recent studies challenge the notion of a neat hierarchical pipeline, wherein each LLM layer serves a fixed linguistic or semantic function. Research by Slobodkin et al. (2021) reveals that context length can shift attention toward different linguistic features across layers. Furthermore, Lioubashevski et al. (2024) demonstrate that a layer's function can change depending on the input. These findings cast doubt on the idea that each layer strictly processes one class of linguistic features. With further research, we hope that a better understanding of the internal, context-dependent processes of linguistic information in LLMs and the human brain will emerge.

Added references:

Belinkov, Y., Màrquez, L., Sajjad, H., Durrani, N., Dalvi, F., & Glass, J. (2018). Evaluating layers of representation in neural machine translation on part-of-speech and semantic tagging tasks. *arXiv preprint arXiv:1801.07772*.

Tenney, I., Das, D., & Pavlick, E. (2019). BERT rediscovers the classical NLP pipeline. *arXiv preprint arXiv:1905.05950*.

Slobodkin, A., Choshen, L., & Abend, O. (2021). Mediators in determining what processing BERT performs first. *arXiv preprint arXiv:2104.06400*.

Lioubashevski, D., Schlank, T., Stanovsky, G., & Goldstein, A. (2024). Looking Beyond The Top-1: Transformers Determine Top Tokens In Order. *arXiv preprint arXiv:2410.20210*